# Seed quality drives grain yield in Ethiopian and Senegalese sorghum: Insights from machine learning

Ezekiel Ahn[1]*, Louis K. Prom[2], Jae Hee Jang[1], Insuck Baek[3], Adama R. Tukuli[2], Seunghyun Lim[1], Seok Min Hong[3,4], Moon S. Kim[3], Lyndel W. Meinhardt[1], Sunchung Park[1], Clint Magill[5]*

1 Sustainable Perennial Crops Laboratory, Agricultural Research Service, Beltsville Agricultural Research Center, United States of America Department of Agriculture, Beltsville, Maryland, United States of America, 2 Insect Control and Cotton Disease Research, Agricultural Research Service, Southern Plains Agricultural Research Center, United States of America Department of Agriculture, College Station, Texas, United States of America, 3 Environmental Microbial and Food Safety Laboratory, Agricultural Research Service, Beltsville Agricultural Research Center, United States of America Department of Agriculture, Beltsville, Maryland, United States of America, 4 Department of Civil Urban Earth and Environmental Engineering, Ulsan National Institute of Science and Technology, UNIST-gil 50, Ulsan, Republic of Korea, 5 Department of Plant Pathology and Microbiology, Texas A&M University, College Station, Texas, United States of America

☯ These authors contributed equally.
* ezekiel.ahn@usda.gov; c-magill@tamu.edu

## Abstract

Accurately predicting grain yield remains a major challenge in sorghum breeding, particularly across genetically and geographically diverse germplasm. To address this, we applied a phenotype-informed machine learning (PIML) framework to analyze nine phenotypic traits in 179 Ethiopian and Senegalese accessions. Using hierarchical clustering and oversampling with ADASYN, we achieved high classification accuracy (0.99) for phenotypic group assignment. Grain yield prediction was most effective with a Neural Boosted model (NTanH(3)NBoost(8)), achieving a mean $R^2$ of 0.36 and RASE (equivalent to RMSE) of 4.87. Feature importance analysis consistently identified seed weight and germination rate as the strongest predictors of grain yield, while disease resistance traits showed limited predictive value. These findings suggest that early selection based on seed quality traits may provide a practical strategy for improving sorghum yield under field conditions, especially in resource-limited environments.

## Introduction

Sorghum (*Sorghum bicolor* (L.) Moench) is a vital food and fodder source in Africa and Asia. Globally, it is more common as animal feed but is gaining attention as a biofuel crop [1]. With over 60 million tonnes produced annually and Africa contributing

**Data availability statement:** The Python scripts used for data preprocessing, model training, and evaluation, along with the yield prediction machine-learning data, are publicly available on GitHub: [Repository name: Seed-quality-drives-grain-yield-in-Ethiopian-and-Senegalese-sorghum-Insights-from-machine-learning] https://github.com/EJSAHN/Seed-quality-drives-grain-yield-in-Ethiopian-and-Senegalese-sorghum-Insights-from-machine-learning. The raw phenotype data analyzed in this study were previously published and can be accessed through references [8] and [9].

**Funding:** This work was supported by the United States Department of Agriculture, Agricultural Research Service (USDA-ARS) In house project # 8042-21220-258-000-D and #3091-22000-040-000-D. The funders had no role in study design, data collection and analysis, decision to publish, or preparation of the manuscript.

**Competing interests:** The authors have declared that no competing interests exist.

~20 million tonnes, it ranks second only to maize on the continent [2]. Sorghum is vulnerable to fungal diseases that reduce yield and quality [3].

Key diseases include anthracnose, grain mold, and rust. *Colletotrichum sublineola*, causing anthracnose, can reduce yields by up to 70% in hot, humid conditions [4]. It spreads easily due to its resilience and dispersal by wind/water [4]. *Fusarium* spp., a common grain mold pathogen, produces mycotoxins like fumonisins, a food safety risk in regions highly dependent on sorghum [5]. Losses range from 30% to 100%, depending on multiple factors [6,7]. Rust (*Puccinia purpurea*), which appears as rust-like spots, can cause yield losses ranging from 13.1% to as high as 65%, depending on the host plant's maturity and susceptibility, as well as environmental conditions [6,7]. Germplasm-based resistance is seen as the most effective control strategy [7]. Resistant genotypes have been identified through screening and breeding efforts [6].

Our previous study evaluated 179 accessions from Ethiopia, Gambia, and Senegal. Traits included yield, seed weight, flowering time, germination, panicle traits, and disease resistance [8,9]. The goal was to identify potential relationships between these traits and disease resistance, which could be a useful asset to sorghum breeding programs. Due to limitations in basic statistical analysis, we applied machine learning to explore complex relationships and identify the best algorithm for trait evaluation. Specifically, this study aims to evaluate the multivariate traits of sorghum using various machine learning algorithms to determine the best approach for analyzing the data. As our previous study states, understanding the genetic diversity and relationships among sorghum accessions is essential for developing improved varieties with enhanced agronomic traits and resilience to biotic and abiotic stresses [8]. Traditional methods for characterizing, classifying, and predicting sorghum traits among germplasms often rely on labor-intensive and time-consuming phenotypic evaluations [10]. While valuable, these approaches may struggle to fully capture the complex, non-linear relationships among multiple traits, the nuances of genotype-by-environment interactions (G×E), and the wealth of information available from high-dimensional genomic, phenomic, and environmental data [11]. Statistical machine learning offers a powerful alternative, extracting patterns from extensive datasets and improving prediction reliability through cross-validation that can be applied to agricultural studies [11]. Standard clustering techniques, such as K-Means, often assume spherical clusters and require prior knowledge of the number of clusters [12]. This limitation motivates the use of machine learning approaches, such as density-based and hierarchical clustering algorithms, which can better capture complex relationships and identify clusters of arbitrary shape [12].

Machine learning is increasingly being applied to address challenges in crop improvement, particularly in leveraging germplasm resources and enhancing stress tolerance and crop breeding [13]. For example, recent studies in maize (*Zea mays*) have utilized machine learning to predict grain yield and stress tolerance indices under both normal and drought conditions, demonstrating the potential for optimizing hybrid selection [12]. In pearl millet, another important staple crop, machine learning has been applied widely, including disease identification, weed management, and land evaluation [14]. Paddy rice smart farming has demonstrated the application

of machine learning and big data across the entire production and post-production chain, including tasks such as yield prediction, disease monitoring, and quality assessment [15]. These studies show how machine learning can accelerate breeding and enhance crop resilience to abiotic and biotic stresses, leading to higher yields.

In general, phenotypic data is collected, refined, and organized, and then used to test ML algorithms. Generally, machine learning algorithms can be categorized into three types: supervised, unsupervised, or reinforcement [16]. In supervised learning, the algorithm is trained on a labeled dataset to yield the desired results [17], whereas in unsupervised learning, the algorithm independently discovers patterns after being given an unlabeled dataset [18]. An algorithm undergoing reinforcement learning receives feedback through rewards and penalties, constantly adjusting its parameters to maximize the rewards it can obtain [13]. Once a machine learning model is made, its performance must be tested. One way this can be done is by splitting the data into a training group and a testing group, in which the algorithm is shaped by the training group and the capability is assessed by the testing group. This method is effective if the data set is on the larger side. The model can also be evaluated using cross-validation; the dataset is split into $n$ groups, and the model is tested $n$ times, with a different subset used as the training group each time and the other $n$-1 subsets used as testing groups. This is the preferred testing method for smaller datasets [13].

While machine learning offers powerful tools for analyzing complex, non-linear, and often high-dimensional data in plant breeding and biotechnology [19], there remains a specific need to evaluate multiple clustering and prediction algorithms within sorghum. To address this, we introduce and apply a Phenotype-Informed Machine Learning (PIML) framework, where we systematically benchmark multiple machine learning models for two key tasks: clustering multivariate phenotypic data and predicting grain yield, a critical agronomic trait. By incorporating both oversampling and undersampling strategies to address class imbalance, we aimed to assess how data distribution influences model performance.

To guide this investigation, we hypothesized that: (1) different machine learning models would exhibit varying performance in clustering sorghum accessions based on multivariate phenotypic data, with some models revealing more biologically meaningful groupings than traditional hierarchical clustering; (2) models capable of capturing non-linear relationships and complex trait interactions would outperform those assuming linearity in predicting grain yield; (3) certain machine learning models would be more effective than others in classifying accessions into pre-defined phenotypic clusters, thereby better reflecting the underlying structure of the data; and (4) seed quality traits—particularly seed weight and germination rate—would encode greater predictive power for yield than disease resistance traits, revealing an underappreciated axis of selection in dryland cereal breeding.

## Materials and methods

### Data preprocessing

The data used in this analysis were gathered from field trials carried out in Isabela, Puerto Rico, during the 2017 and 2018 growing seasons [8,9]. The study evaluated a total of 201 accessions. This set was composed of 179 diverse sorghum accessions originating from Ethiopia, Gambia, and Senegal, along with 22 previously characterized control lines used for comparison, covering various phenotypic traits such as disease resistance (Anthracnose, Rust, Grain mold), seed quality (Seed weight, Germination rate), growth (Panicle height, Panicle length), and yield (Grain yield). These accessions, provided by the USDA-ARS Plant Genetic Resources Conservation Unit, Griffin, Georgia were selected to evaluate their potential as genetic sources for breeding, specifically for enhancing resistance to biotic stress and improving other key agronomic traits [8,9]. Furthermore, the dataset contains Julian days for flowering dates. The data were imported into JMP Pro 17 for initial processing, and missing values were handled using Multivariate Normal Imputation with least squares prediction. This method was selected for its effectiveness in preserving the covariance structure and interrelationships among the phenotypic traits in the dataset, thereby providing a more robust imputation than simpler methods, such as mean replacement. Prior to imputation, no accessions were filtered based on the extent of missing data [20]. Disease

resistance scores were recorded on a scale of 1–5, following the standard evaluation protocols detailed in our previous work with this collection [8,9].

## Statistical analysis

*t*-tests were conducted using JMP Pro 17 [21] to compare phenotypic traits between accessions from Ethiopia and Senegal. Accessions from Gambia (n = 12) were excluded from this specific statistical comparison due to the small sample size. Principal component analysis (PCA) was conducted to explore the phenotypic diversity and trait relationships. To identify groups within the data, hierarchical clustering was applied to all traits using Ward's linkage method. This analysis revealed four distinct clusters of accessions (Fig 1). The selection of four clusters was based on a visual inspection of the dendrogram, which provided a balance between achieving clear separation and maintaining a practical number of groups for the subsequent machine learning classification analysis.

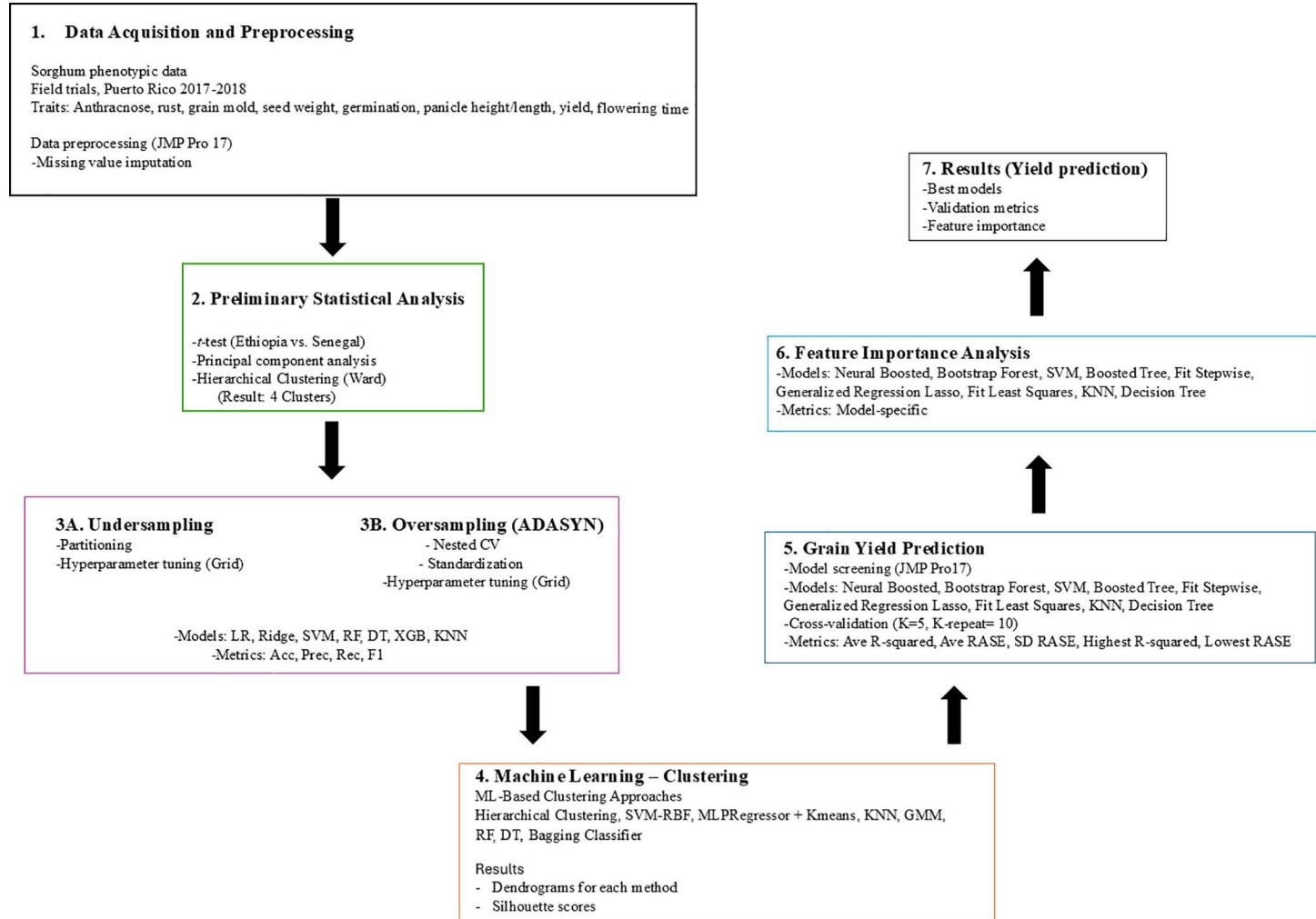

**Fig 1. Methodological flowchart outlining the study's workflow.** The process begins with data acquisition from field trials. It proceeds through preprocessing, preliminary statistical analysis (including PCA and hierarchical clustering), and three primary machine learning tasks: cluster classification (using both undersampling and ADASYN oversampling), machine learning-based clustering, and grain yield prediction. Key algorithms and evaluation metrics are indicated within each stage.

## Machine learning analysis

### Overall workflow.

**Cluster classification. Random undersampling:** We created a balanced dataset of 248 data points by first identifying the size of the smallest of the four clusters (n = 62) and then randomly selecting 62 accessions from each of the four clusters to balance the dataset (62 samples × 4 clusters = 248 total points). Features were standardized using a StandardScaler. The dataset was partitioned into training (60%), validation (20%), and testing (20%) sets using stratified splits. Hyperparameter tuning was performed using GridSearchCV with 5-fold stratified cross-validation (StratifiedKFold, n_splits = 5, random_state = 42, n_jobs = −1) on the training set, and f1_macro was used as scoring. The macro-averaged F1-score was chosen as the scoring metric because it calculates the harmonic mean of precision and recall for each class independently, then averages these values. This approach treats all classes equally, making it a robust metric for multi-class classification tasks with class imbalance. The following models were used: Logistic Regression [22], Ridge Classifier [23], Support Vector Machines with Radial basis function kernel (SVM-RBF) [24], Random Forest [25], Decision Tree [26], Boosted Tree (XGBoost) [27], and K-Nearest Neighbors (KNN) [28]. These models were first evaluated using their default hyperparameters during the screening process, before being specifically tuned with GridSearchCV. These models represent different underlying approaches, including linear models (Logistic Regression, Ridge Classifier), kernel methods (SVM-RBF), tree-based ensembles (Random Forest, Boosted Tree/XGBoost), an instance-based method (KNN), and a Decision Tree, allowing for comparison across various levels of complexity and assumptions Table 1. The best-performing model for each algorithm, based on the highest validation accuracy from the GridSearchCV internal cross-validation, was then evaluated on the held-out validation set [29]. Independently, to assess performance variability, a separate 5-fold stratified cross-validation (StratifiedKFold, n_splits = 5, shuffle = True, random_state = 42) was performed on the training data. The mean and standard deviation of the accuracy across these folds are reported. Finally, each best model identified was evaluated on the test set to measure generalization [29].

**Adaptive Synthetic (ADASYN) sampling approach:** To address class imbalance while retaining original data, ADASYN was implemented using imblearn (v0.10.1) [30]. We selected ADASYN not only because, as an oversampling technique, it preserves all original data points (unlike undersampling), but also because it adaptively generates more synthetic data for minority class samples that are harder to learn, potentially leading to better model performance compared to simpler oversampling methods like SMOTE which generates samples more uniformly. With random_state = 42 for reproducibility,

**Table 1. Summary of machine learning models used for cluster classification.**

| Model name | Type | Brief description |
|---|---|---|
| Logistic Regression | Linear Model | A statistical model that predicts the probability of a categorical outcome. |
| Ridge Classifier | Linear Model | A linear classification model with L2 regularization that helps prevent overfitting by penalizing large coefficients. |
| SVM (RBF Kernel) | Kernel-based, non-linear | A kernel-based method that maps data to a higher-dimensional space using the RBF kernel to find a separating hyperplane. |
| Random Forest | Ensemble (Bagging) | An ensemble of decision trees trained on bootstrapped samples (Bagging), reducing variance and improving prediction accuracy. |
| Boosted Tree (XGBoost) | Ensemble (Boosting) | A gradient boosting method that builds trees sequentially, where each tree corrects the errors of the previous ones. XGBoost is a high-performance implementation of this approach. |
| K-Nearest Neighbors | Instance-based | A non-parametric method that classifies a data point based on its neighbors. |
| Decision Tree | Single tree model | A simple, interpretable model that splits data based on feature values. |

ADASYN was applied to the raw data, after loading data and before any data splitting or feature scaling, using adasyn. fit_resample(X, y). This produced a balanced, oversampled dataset of 5379 data points. A 5-fold stratified cross-validation (StratifiedKFold, n_splits = 5, random_state = 42, shuffle = True) was applied to the oversampled data. Within each fold, feature standardization was performed using StandardScaler (scikit-learn), fit only to the training data and then applied to the testing data, preventing data leakage. Hyperparameter tuning was conducted within each outer fold using GridSearchCV (scikit-learn), which itself employed a 3-fold stratified cross-validation (StratifiedKFold, n_splits = 3, n_jobs = −1). The macro-averaged F1-score was used for optimization.

**Machine learning-based clustering of sorghum accessions.** To explore the relationships among sorghum accessions based on phenotypic traits, dendrograms were generated using several machine learning and statistical methods, with hierarchical clustering as a common visualization tool. First, a baseline dendrogram was created by applying hierarchical clustering with Ward's linkage directly to the standardized trait data. Next, several machine learning models were trained, and their outputs were used to inform the hierarchical clustering. An SVM-RBF kernel was trained to classify accessions [31]. The distances of each sample to the decision boundary, obtained via the decision_function method, were then used as input for hierarchical clustering with Ward's linkage. A Random Forest classifier was also trained, and a proximity matrix was derived [25]. If out-of-bag predictions were available (via oob_decision_function_), they were used to construct the proximity matrix; otherwise, a custom function (proximity_matrix_from_trees) calculated the proportion of trees where two samples ended up in the same terminal node. This proximity matrix was converted to a distance matrix and used for hierarchical clustering with Ward's linkage. A Multilayer Perceptron Regressor (MLPRegressor) [32], with two hidden layers (sizes 10 and 5), ReLU activation, the Adam solver, and 1000 maximum iterations, was employed as an encoder to reduce the dimensionality of the data [32]. Based on a preliminary analysis using the elbow method to evaluate the within-cluster sum of squares on the MLP-encoded data, K-Means clustering with k = 3 was then applied. Hierarchical clustering (Ward's linkage) was applied to the distances between data points in the reduced-dimensional space created by the MLPRegressor. A KNN classifier was also trained [28]; the maximum distance to the k-nearest neighbors for each sample was calculated. This distance, combined with a standard Euclidean distance matrix calculated from the standardized data, was fed into hierarchical clustering with Ward's linkage. A Gaussian Mixture Model (GMM) with three components was fit directly to the original scaled data [33]. The Mahalanobis distances between each sample and the means of the GMM-determined clusters were used in Ward's linkage hierarchical clustering. Additionally, a Decision Tree classifier was trained [26]. Finally, a Bagging Classifier [34], using Decision Trees as base estimators (10 estimators, with a fixed random state), was trained. A proximity matrix, representing the proportion of trees where two samples shared a terminal node (calculated using the proximity_matrix_from_trees function), was derived from the Bagging Classifier. Ward's linkage hierarchical clustering was applied to the distance matrix derived from this proximity matrix. For all methods, hierarchical clustering was performed using the linkage function from the scipy.cluster.hierarchy module, and dendrograms were generated using the dendrogram function. While other internal validation indices, such as the Davies-Bouldin index were considered, the silhouette score, calculated using the silhouette_score function from sklearn.metrics, was used to assess the quality of cluster separation for each method, where applicable.

## Grain yield prediction: model screening and selection

For grain yield prediction and feature importance analysis, JMP Pro 17's "Model Screening" function, which applies all available prediction methods in the software, was employed to compare various machine learning models with 10 repeats of 5-fold cross-validation (random seed = 1), using default settings [21]. This screening process generated multiple models, including Neural Boosted model, for which the specific architecture identified by JMP Pro is represented by the notation "NTanH(3)NBoost(8)"; this indicates a neural network with a single hidden layer containing 3 neurons utilizing the TanH (Hyperbolic Tangent) activation function, subsequently boosted using an ensemble method involving 8 boosting iterations.

## Results

### Principal component analysis and hierarchical clustering analysis

PCA was employed to investigate phenotypic diversity and trait correlations among Ethiopian and Senegalese sorghum accessions, with additional accessions from other countries included. The PCA biplot (Fig 2a) visualizes the distribution of these accessions based on nine phenotypic traits, with the first two principal components explaining 47.6% of the total variance. While this indicates that additional components capture a significant portion of the remaining variance, the first two PCs are presented here as they effectively illustrate the primary patterns of geographical clustering and the most dominant trait associations within the dataset. The PCA plot reveals a close separation between the average values for Ethiopia and Senegal, with Gambia and Sudan positioned between them. Senegal clusters near Gambia, followed by Sudan and then Ethiopia, mirroring their geographical proximity. A focused PCA biplot on specific traits (Figs 2b and 2c) shows distinct clustering: Seed weight and grain yield form one cluster, while germination rate, panicle height, flowering date, and rust resistance are somewhat linked. Grain mold and anthracnose resistance appear distinctly separate from other traits. The loading plot (Fig 2b) clarified the direction and nature of these trait correlations with the principal components. For example, the vectors for 'Seed weight' and 'Grain yield' are closely aligned, indicating a strong positive correlation between them, and both contribute significantly to PC2. 'Panicle height' and 'Julian days' are positively correlated with PC1, while 'Grain mold' resistance is strongly negatively correlated with PC1. 'Anthracnose' and 'Grain yield', with vectors at around a 90-degree angle to each other, show little to no correlation. The length of each vector indicates the strength of that trait's contribution to the variance explained by the principal components.

A comparative analysis revealed significant differences in most phenotypic traits between Ethiopian and Senegalese sorghum accessions (Table 2). Except for grain yield, all traits differed significantly between the two origins. Ethiopian accessions were more susceptible to anthracnose ($2.29 \pm 0.017$ (standard error) vs. $2.11 \pm 0.007$) and grain mold ($2.6 \pm 0.037$ vs. $3.03 \pm 0.033$), while Senegalese lines showed higher susceptibility to rust ($3.32 \pm 0.012$ vs. $3.49 \pm 0.017$). Although seed weight was slightly higher in Ethiopian accessions, Senegalese accessions exhibited greater germination rates, panicle lengths, plant heights, and days to flowering (Julian days). These significant phenotypic differences likely reflect divergent selection pressures and adaptations to the distinct agro-ecological zones of Ethiopia (East Africa) and Senegal (West Africa). The higher rust susceptibility in Senegalese accessions compared to Ethiopian accessions may be due to the prevalence of different pathogen races. The greater resistance to grain mold in Senegalese accessions could be an adaptation to the higher humidity conditions common in parts of that region during the growing season. The observed differences in agronomic traits, such as panicle height and flowering time, further indicate adaptations to local farming systems and environmental cues.

The table depicts the distributions of nine phenotypic traits for the collection of Ethiopian and Senegalese accessions. Standard errors are provided alongside the average values for clarity. Except for grain yield, all other traits showed significant differences, with $t$-test results being statistically significant. * indicates significance at $p < 0.05$.

Hierarchical clustering analysis (Fig 3a) identified four distinct clusters within the sorghum accessions based on all nine phenotypic traits. The trait dendrogram (Fig 3b) further revealed the relationships between these traits, showing distinct groupings of panicle height and Julian days (flowering time). Rust resistance clustered closely with germination rate and panicle length, while anthracnose resistance was more closely associated with seed weight and grain yield. Grain mold resistance remained distinct from the other traits.

### Machine learning-based classification

The dataset was analyzed using hierarchical clustering, identifying four distinct clusters (Fig 3a). We attempted to classify individual accessions directly based on all nine traits. However, preliminary analyses using a suite of machine learning algorithms (including SVM, Random Forest, and KNN) resulted in low accuracy (below 10% in all tested models). This

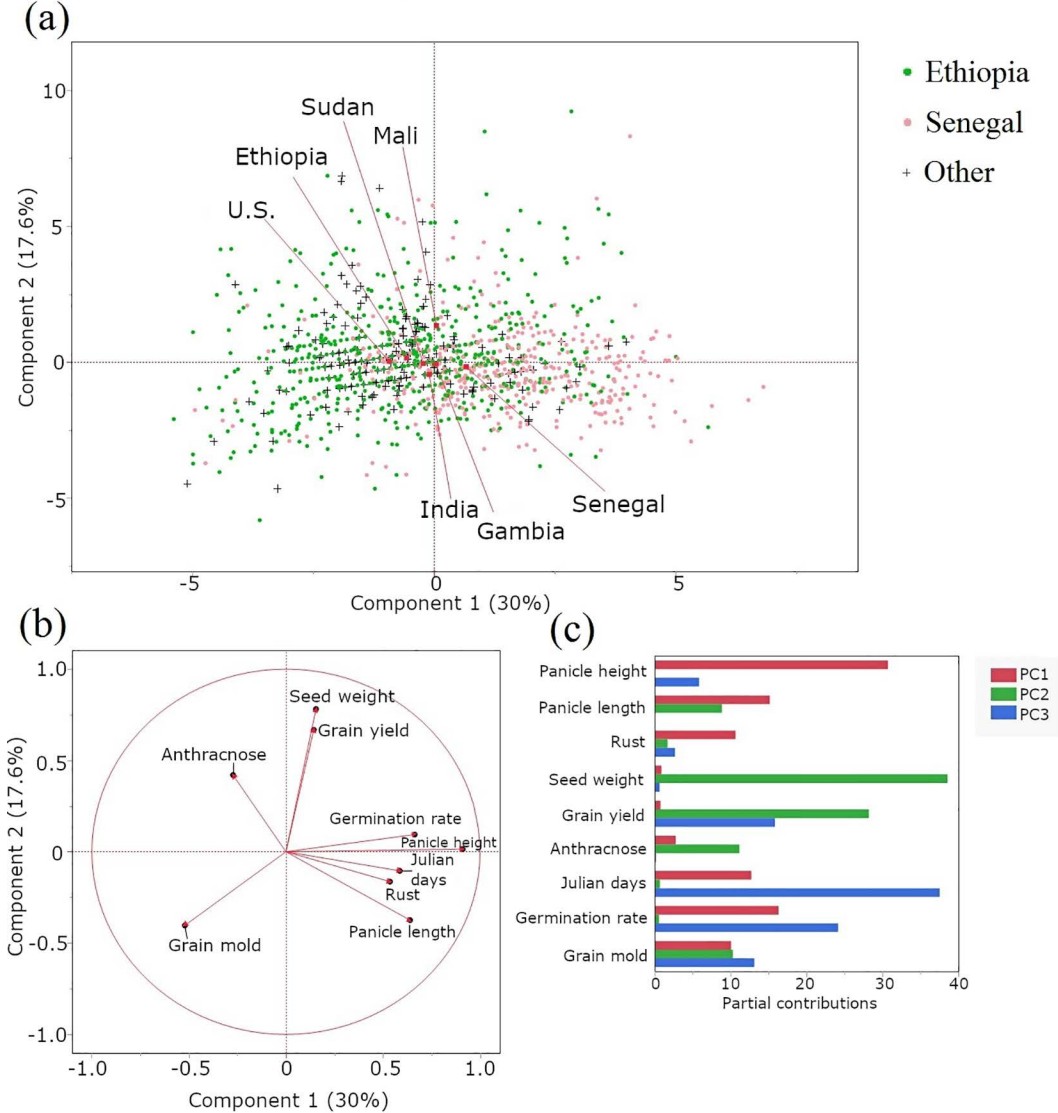

**Fig 2. Principal component analysis of Ethiopian and Senegalese sorghum accessions. (a)** Displaying a scatter plot of sorghum accessions, showcasing their distribution based on the first two principal components. The accessions are color-coded by their geographic origin (Ethiopia, Senegal, or others), providing a visual representation of the overall diversity in nine traits and the relationships among the accessions. The red points on the plot represent the average location of each country. **(b)** Loading plot showing the contribution of each trait to the first two principal components. The direction and length of the arrows indicate the strength and direction of the correlation between each trait and the principal components. Traits closely clustered together are positively correlated. **(c)** Bar plot illustrating the partial contributions of each trait to the first three principal components, identifying the traits that most strongly influence the variation captured by each principal component.

poor performance was expected and can be attributed to two main factors: a severe class imbalance, where each of the 179 accessions represented its own class with only a single sample, and a challenging feature-to-sample ratio. With only nine phenotypic traits to distinguish between 179 unique classes, the models were unable to learn generalizable patterns and were prone to overfitting. To address this, we adopted a two-step approach: first, hierarchical clustering grouped accessions into four distinct clusters (Fig 3a). Second, we evaluated seven machine-learning algorithms to classify accessions into these predefined clusters. To mitigate the class imbalance, we employed both undersampling and oversampling

**Table 2. Comparison of phenotypic traits between Ethiopian and Senegalese accessions.**

| Category | Trait | Ethiopian Accessions | Senegalese Accessions | *p*-value |
|---|---|---|---|---|
| Disease Resistance | Anthracnose (1–5 score) | 2.29±0.01 | 2.11±0.01 | <0.0001* |
| | Rust (1–5 score) | 2.6±0.04 | 3.03±0.03 | <0.0001* |
| | Grain Mold (1–5 score) | 3.49±0.01 | 3.32±0.01 | <0.0001* |
| Seed Quality | Seed Weight (g) | 2.13±0.01 | 2.08±0.01 | <0.0001* |
| | Germination Rate (%) | 62.75±0.45 | 69.64±0.44 | 0.001* |
| Agronomic Traits | Panicle Height (cm) | 161.14±1.47 | 194.2±1.42 | <0.0001* |
| | Panicle Length (cm) | 22.03±0.24 | 28.24±0.23 | <0.0001* |
| | Julian Days (to flowering) | 117.03±0.17 | 118.58±0.16 | <0.0001* |
| | Grain Yield (g/3 panicles) | 16.12±0.19 | 16.37±0.19 | 0.33 |

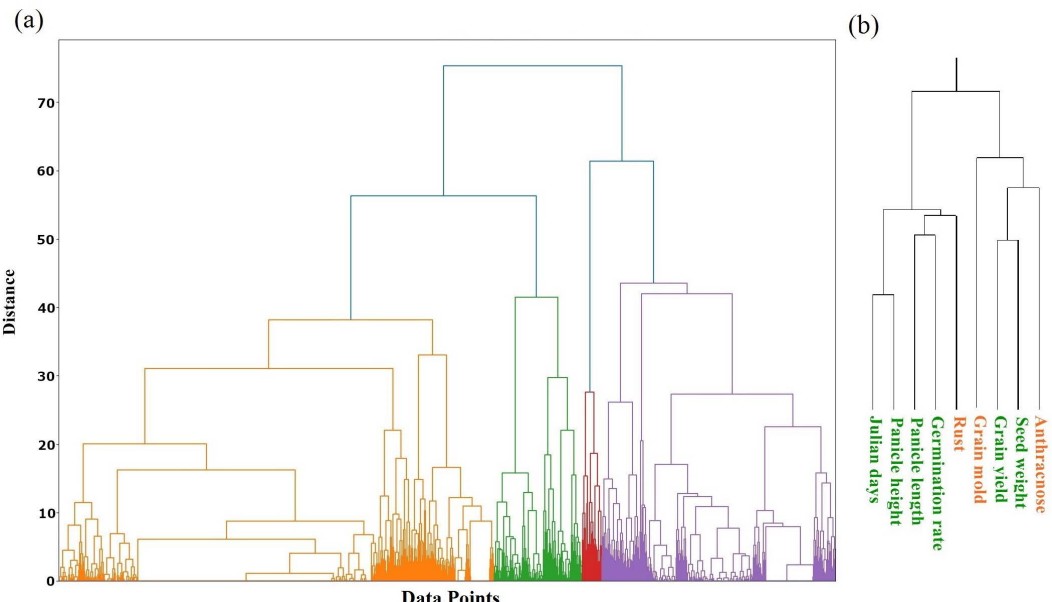

**Fig 3. Hierarchical Clustering Analysis of Sorghum Accessions and Traits. (a)** Dendrogram illustrating the hierarchical clustering of sorghum accessions based on phenotypic traits. The dendrogram was constructed using the DIANA (Divisive Analysis Clustering) algorithm. **(b)** Hierarchical clustering of phenotypic traits. The dendrogram reveals the relative similarity between traits, with closely related traits clustered together.

(ADASYN) techniques. Results for the seven machine learning algorithms using the undersampled dataset are shown (Table 3). Hyperparameter tuning was performed using 5-fold stratified cross-validation. The SVM with an RBF kernel achieved the highest validation accuracy (0.94), followed closely by Logistic Regression (0.92), Boosted Trees, K-Nearest Neighbors (both 0.90), and Decision Tree (0.90). The Ridge Classifier had the lowest validation accuracy (0.82) but still performed reasonably well. On the independent test set, Logistic Regression achieved the highest accuracy (0.96), followed by KNN, Random Forest, and Boosted Tree (all 0.92). The SVM, while performing best on the validation set, had a slightly lower test accuracy (0.90). The Decision Tree had the lowest test accuracy (0.84) of these models.

For the ADASYN-oversampled dataset, hyperparameter tuning was conducted using a nested cross-validation approach (3-fold inner loop for tuning, 5-fold outer loop for evaluation) (Table 4). Due to the effectiveness of ADASYN in creating a balanced dataset, all tested models achieved near-perfect performance (accuracy, precision, recall, and

**Table 3. Performance of machine learning models on the cluster classification task after hyperparameter tuning with 5-fold stratified cross-validation, using an undersampled dataset.**

| Model | Validation Accuracy | Validation Precision (Macro Avg) | Validation Recall (Macro Avg) | Validation F1-score (Macro Avg) | Test Accuracy | Test Precision (Macro Avg) | Test Recall (Macro Avg) | Test F1-score (Macro Avg) |
|---|---|---|---|---|---|---|---|---|
| Logistic Regression | 0.92 | 0.92 | 0.91 | 0.91 | 0.96 | 0.97 | 0.97 | 0.97 |
| Ridge Classifier | 0.82 | 0.86 | 0.84 | 0.81 | 0.76 | 0.81 | 0.8 | 0.76 |
| SVM (RBF) | 0.94 | 0.94 | 0.93 | 0.94 | 0.9 | 0.9 | 0.92 | 0.91 |
| Random Forest | 0.88 | 0.87 | 0.88 | 0.87 | 0.92 | 0.93 | 0.93 | 0.92 |
| Decision Tree | 0.9 | 0.9 | 0.9 | 0.9 | 0.84 | 0.86 | 0.86 | 0.85 |
| Boosted Tree | 0.9 | 0.9 | 0.89 | 0.9 | 0.92 | 0.93 | 0.93 | 0.92 |
| K-Nearest Neighbors | 0.9 | 0.91 | 0.9 | 0.9 | 0.92 | 0.93 | 0.93 | 0.92 |

Validation set performance and test set performance (accuracy, macro-averaged precision, recall, and F1-score) are reported.

F1-score of 0.99). However, these scores should be interpreted with caution, as they are likely inflated due to the application of oversampling to the entire dataset prior to cross-validation splitting, which can lead to information leakage.

Due to the effectiveness of ADASYN in balancing the classes and abundance of data points, all tested models achieved near-perfect performance, choosing hyperparameters that were less critical in this specific case. All performance metrics (accuracy, macro-averaged precision, recall, and F1-score) are calculated across all folds of the 5-fold cross-validation.

## Machine learning-based clustering of sorghum accessions

We investigated potential differences in clustering patterns using various machine learning algorithms compared to traditional hierarchical clustering analysis (Fig 4a). To quantify the quality of the clusterings produced by each method, we calculated the silhouette score, which measures how similar an object is to its own cluster compared to other clusters. The dendrograms generated by the different models revealed distinct clustering structures within the sorghum accession dataset (Figs 4b–4h). The MLPRegressor (Neural Network) model, combined with K-Means clustering, produced a dendrogram (Fig 4h) with well-defined clusters, reflected in its relatively high silhouette score of 0.317. Similarly, the GMM (Fig 4c) also showed a relatively high silhouette score of 0.311, potentially reflecting the presence of inherent groupings within the data, consistent with the hierarchical clustering. The SVM-RBF model (Fig 4f) produced a dendrogram with two primary clusters like GMM, although the separation was less pronounced than with the MLPRegressor+K-Means. This

**Table 4. Performance metrics for machine learning models classifying sorghum accessions into four clusters on an ADASYN-oversampled dataset.**

| Model | Accuracy | Precision (Macro Avg) | Recall (Macro Avg) | F1-score (Macro Avg) |
|---|---|---|---|---|
| Logistic Regression | 0.99 | 0.99 | 0.99 | 0.99 |
| Ridge Classifier | 0.99 | 0.99 | 0.99 | 0.99 |
| SVM | 0.99 | 0.99 | 0.99 | 0.99 |
| Random Forest | 0.99 | 0.99 | 0.99 | 0.99 |
| Decision Tree | 0.99 | 0.99 | 0.99 | 0.99 |
| Boosted Tree | 0.99 | 0.99 | 0.99 | 0.99 |
| KNN | 0.99 | 0.99 | 0.99 | 0.99 |

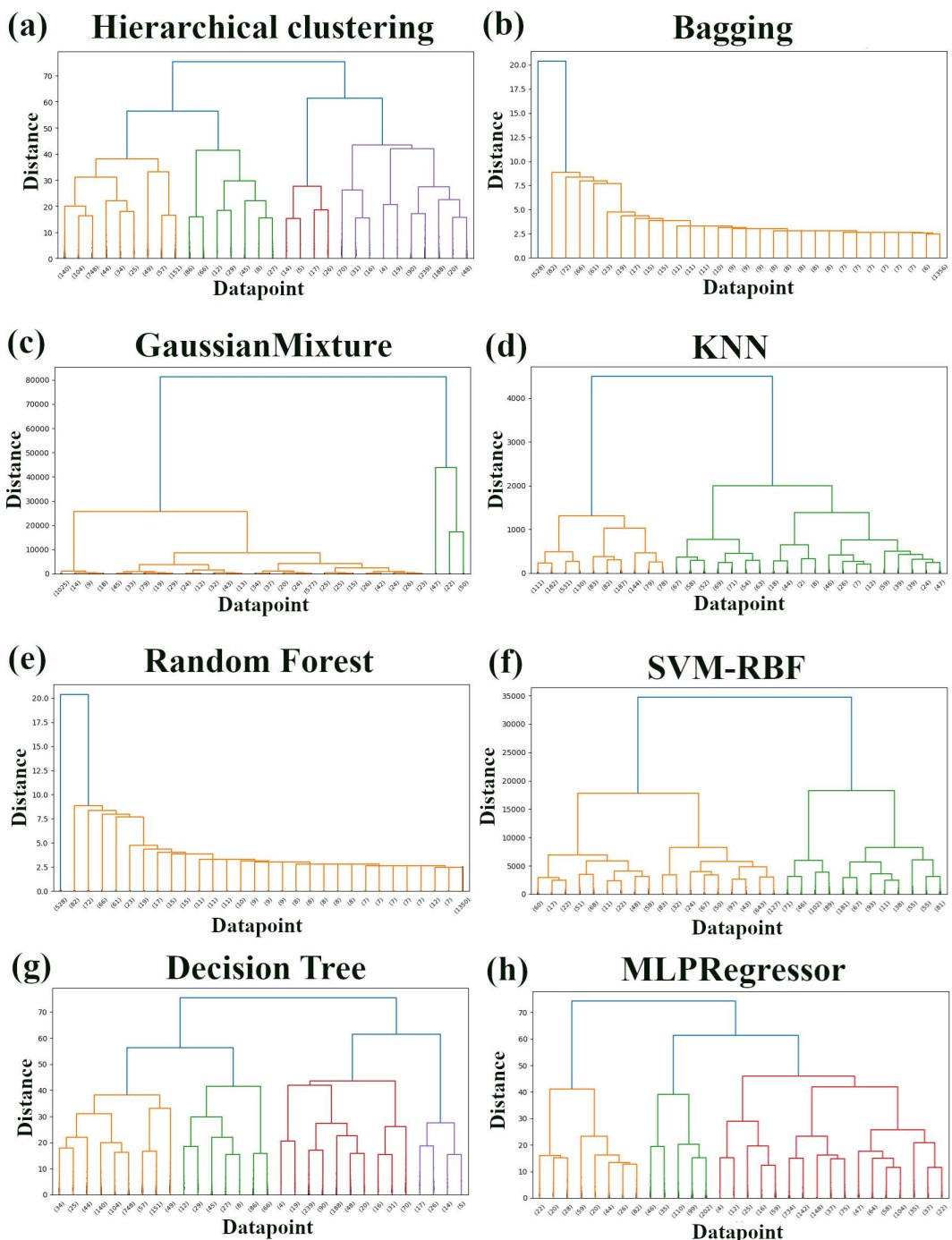

**Fig 4. Comparative analysis of Ethiopian and Senegalese accessions clustering using various machine learning and statistical techniques.** Dendrograms were generated using different methods that were applied to phenotypic trait data. Numbers at the end of branches indicate the number of samples within each cluster. **(a)** Ward's linkage based on non-machine learning clustering. **(b)** Bagging classifier. **(c)** Gaussian Mixture. **(d)** KNN. **(e)** Random Forest. **(f)** SVM-RBF. **(g)** Decision Tree. **(h)** MLP Regressor (Neural Network). The y-axis represents the distance or dissimilarity between clusters or data points. All dendrograms were constructed using hierarchical clustering with Ward's linkage.

is reflected in its moderate silhouette score of 0.234. In contrast, Random Forest (Fig 4e), Bagging Classifier (Fig 4b), Decision Tree (Fig 4g), and KNN-based (Fig 4d) methods exhibited significantly lower silhouette scores of 0.118, 0.127, 0.122, and 0.087, respectively. The dendrograms for these methods visually appear as a more continuous distribution of accessions, or a chaining pattern for Bagging and Random Forest, indicating that these algorithms, in this context, do not identify strong, distinct clusters in the data based on the evaluated traits in the same way as hierarchical clustering or the best-performing clustering algorithms.

**Evaluation of grain yield prediction models**

To evaluate the effectiveness of various machine learning models in predicting grain yield, a critical trait in sorghum breeding, model performance was assessed using several metrics: the average R-squared (coefficient of determination, indicating the proportion of variance in grain yield explained by the model), the average Root Absolute Standard Error (RASE, representing the prediction error of grain yield), the standard deviation of RASE (SD RASE), the highest R-squared, and the lowest RASE, all calculated across validation datasets (Table 5). Among the models tested, the Neural Boosted model (NTanH(3)NBoost(8)) exhibited the highest average R-squared value in the validation set (0.3597), indicating the best overall predictive performance for grain yield. Bootstrap Forest followed with an average R-squared of 0.3327, and SVM-RBF achieved an average R-squared of 0.2917. The remaining models, including Boosted Tree, Fit Stepwise, Generalized Regression Lasso, Fit Least Squares, KNN, Decision Tree, and others, showed progressively lower average R-squared values, ranging from 0.2876 down to 0.1777. Examining the average RASE, which quantifies the prediction error, the Neural Boosted model also performed favorably with a value of 4.873. Lower RASE values indicate better prediction accuracy. Bootstrap Forest and SVM-RBF had average RASE values of 4.9657 and 5.1306, respectively. The Decision Tree model had the highest average RASE at 5.5271, indicating the largest prediction errors among the models tested.

The highest R-squared values achieved by each model across validation sets indicate their optimal performance potential. Neural Boosted, Bootstrap Forest, and SVM reached maximum R-squared values of 0.4784, 0.4737, and 0.4331, respectively. Boosted Tree has the smallest SD RASE value, followed by Neural Boosted, while Decision Tree, Generalized Regression Lasso, and KNN have large values, reflecting the consistency of model performance. Training set performance was considerably higher than validation set performance across several models. For instance, Boosted Tree achieved R-squared values 0.9 on the training set, Bootstrap Forest around 0.75, and both SVM and Neural Boosted around 0.50. This large discrepancy between training and validation performance strongly suggests that some models, particularly Boosted Tree, may be overfitting the training data, learning noise, and specific patterns that do not generalize well. Therefore, the focus here is on the validation set performance, which provides a more realistic and reliable estimate of a model's ability to predict grain yield in new, unseen sorghum accessions.

To understand which phenotypic traits most strongly influence grain yield prediction, feature importance scores were calculated using four different machine learning models: Neural Boosted, SVM-RBF, Bootstrap Forest, and Boosted Tree (Table 6). Across these models, seed weight consistently emerged as a highly influential predictor. In the Neural Boosted model, seed weight had a total effect 0.599. Similarly, the SVM-RBF model assigned a total effect 0.617 to seed weight. For the Bootstrap Forest and Boosted Tree models, seed weight had importance scores of 0.3276 and 0.3987, respectively.

Beyond seed weight, germination rate, panicle length, and plant height were also identified as influential predictors, although their relative importance varied across models. In the Neural Boosted model, panicle height had a total effect of 0.239 and a germination rate of 0.092. In contrast, pathogen resistance traits (anthracnose, rust, and grain mold) generally exhibited lower importance scores. This suggests that while disease resistance is important for overall crop health, the variation in grain yield within this specific dataset was more strongly driven by inherent seed and plant characteristics than by differences in resistance to these particular pathogens.

**Table 5. Performance comparison of machine learning models for grain yield prediction in sorghum accessions.**

| Method | Average R square | Average RASE | SD RASE | Highest R square | Lowest RASE |
|---|---|---|---|---|---|
| Neural Boosted | 0.3597 | 4.873 | 0.65 | 0.4784 | 4.4814 |
| Bootstrap Forest | 0.3327 | 4.9657 | 0.66 | 0.4737 | 3.8592 |
| Support Vector Machines | 0.2917 | 5.1306 | 0.71 | 0.4331 | 3.8365 |
| Boosted Tree | 0.2876 | 5.1154 | 0.57 | 0.4787 | 4.4839 |
| Fit Stepwise | 0.2746 | 5.1852 | 0.68 | 0.3966 | 4.0797 |
| Generalized Regression Lasso | 0.2713 | 5.2003 | 0.72 | 0.3966 | 4.0797 |
| Fit Least Squares | 0.2708 | 5.1975 | 0.68 | 0.3966 | 4.0797 |
| K Nearest Neighbors | 0.2598 | 5.235 | 0.70 | 0.4049 | 4.7866 |
| Decision Tree | 0.1777 | 5.5271 | 0.75 | 0.2892 | 4.4278 |

Models were evaluated on validation datasets using the following metrics: average R-squared (coefficient of determination, higher values indicate better fit), average RASE (lower values indicate better accuracy), standard deviation of SD RASE, highest R-squared achieved, and lowest RASE achieved. The models tested include Neural Boosted, Bootstrap Forest, SVM-RBF, Boosted Tree, Fit Stepwise, Generalized Regression Lasso, Fit Least Squares, KNN, and Decision Tree.

## Discussion

### Phenotypic diversity and geographical patterns

Sorghum, a crucial crop in the drier tropics, is vital for food security, particularly in Ethiopia, where it sustains millions [9,35]. However, its production is constrained by biotic stresses like fungal diseases and abiotic factors like drought, leading to significant yield losses [36]. Using the SARRA-O crop model, Faye et al. projected that rising temperatures could cause significant yield declines of up to 30–40% for short-cycle millet and sorghum varieties in Niakhar, Senegal, by 2050 [37].

While past studies have demonstrated significant differences in sorghum characteristics and the relationships between its morphology and pathogen resistance [8,12], additional investigation is needed to translate these findings into superior grain yield [38]. To overcome essential challenges in sorghum breeding, this research trained and validated effective machine learning models to forecast grain yield and pinpointed crucial phenotypic traits that influence yield potential. We further advanced germplasm characterization by systematically comparing multiple machine learning approaches for phenotype-based clustering, providing a framework for more efficient sorghum improvement.

To gain a foundational understanding of phenotypic diversity and the relationships among the examined traits, we conducted PCA analysis and *t*-tests to statistically compare accessions from Ethiopia and Senegal. The PCA analysis (Fig 2a) clearly distinguishes between the accessions from Ethiopia and those from Senegal. Additionally, fewer accessions from Gambia and Sudan fall in between, reflecting the geographical distribution of these countries. Strikingly, Senegal accessions clustered near Gambia, while Sudan accessions clustered closer to Ethiopia, echoing the geographical layout of the four countries. This spatial distribution suggests a potential influence of geographical origin on phenotypic traits, likely reflecting adaptation to local environments or historical patterns of gene flow. For instance, accessions from Senegal were found to be more susceptible to rust than accessions from Ethiopia. The higher prevalence of rust in East Africa [39] may explain the increased resistance of Ethiopian sorghum to this disease.

### Machine learning for germplasm classification

Our study's application of a PIML framework not only provided robust predictions but also yielded deeper insights into the phenotypic structure of the germplasm. The two-step process, beginning with unsupervised clustering before moving to supervised classification, proved highly effective. While hierarchical clustering based on all nine traits identified four

**Table 6. Feature importance scores indicating the relative contribution of different traits to grain yield prediction in sorghum.**

**Neural Boosted**

| Traits | Main effect | Total effect | Importance |
|---|---|---|---|
| Seed weight | 0.56 | 0.599 | 6 |
| Panicle height | 0.21 | 0.239 | 2 |
| Germination rate | 0.064 | 0.092 | 1 |
| Grain mold | 0.029 | 0.052 | 0 |
| Rust | 0.025 | 0.041 | 0 |
| Panicle length | 0.014 | 0.031 | 0 |
| Julian days | 0.014 | 0.028 | 0 |
| Anthracnose | 0.009 | 0.019 | 0 |

**Bootstrap Forest**

| Traits | Sum of square | Portion | Importance |
|---|---|---|---|
| Seed weight | 10395.1634 | 0.3276 | 18 |
| Germination rate | 4799.4103 | 0.1512 | 8 |
| Julian days | 3837.22564 | 0.1209 | 6 |
| Rust | 3331.06732 | 0.105 | 5 |
| Panicle height | 3168.94918 | 0.0999 | 5 |
| Panicle length | 2587.32949 | 0.0815 | 4 |
| Grain mold | 2537.70953 | 0.08 | 4 |
| Anthracnose | 1078.55728 | 0.034 | 1 |

**SVM-RBF**

| Traits | Main effect | Total effect | Importance |
|---|---|---|---|
| Seed weight | 0.211 | 0.617 | 6 |
| Panicle length | 0.029 | 0.367 | 4 |
| Panicle height | 0.066 | 0.293 | 3 |
| Grain mold | 0.041 | 0.251 | 2 |
| Germination rate | 0.018 | 0.15 | 1 |
| Anthracnose | 0.019 | 0.14 | 1 |
| Julian days | 0.018 | 0.111 | 1 |
| Rust | 0.022 | 0.092 | 1 |

**Boosted Tree**

| Traits | Sum of square | Portion | Importance |
|---|---|---|---|
| Seed weight | 109409.754 | 0.3987 | 18 |
| Germination rate | 56006.7156 | 0.2041 | 9 |
| Julian days | 41134.3698 | 0.1499 | 6 |
| Grain mold | 23474.5505 | 0.0856 | 3 |
| Rust | 16902.2966 | 0.0616 | 2 |
| Panicle height | 14290.9634 | 0.0521 | 2 |
| Panicle length | 6602.28891 | 0.0241 | 1 |
| Anthracnose | 6571.50613 | 0.0239 | 1 |

Scores were obtained from four machine learning models: Neural Boosted, Bootstrap Forest, SVM-RBF, and Boosted Tree. The specific metric used to quantify importance varies by model

distinct groups (Fig 3a), interpreting their precise biological meaning requires caution, given the modest silhouette scores, which indicate weak separation. However, we can speculate on potential underlying factors by considering the trait relationships revealed in the PCA (Fig 2b, 2c) and trait dendrogram (Fig 3b), alongside the known differences between Ethiopian and Senegalese accessions (Table 2).

For instance, we can speculate that one cluster might group accessions exhibiting traits potentially associated with Ethiopian origin, such as earlier flowering (Julian days) and higher susceptibility to grain mold. These traits clustered somewhat separately in our analysis. Another cluster could potentially represent accessions more typical of Senegalese origin, characterized by later flowering, taller plants (Panicle height), higher germination rates, and perhaps higher susceptibility to rust. These traits showed associations in Fig 3b. Other clusters may represent combinations emphasizing high seed weight and yield potential (which clustered with anthracnose resistance) or specific adaptations not fully captured by origin alone. This interpretation remains speculative and primarily serves to generate hypotheses about the complex interplay between adaptation, growth habit, and geographical origin. Therefore, the goal of this exploratory clustering was to reveal potential patterns that warrant further investigation, which would ideally be confirmed by integrating genotypic data to understand the underlying genetic basis of these groupings.

Machine learning is increasingly being applied to address challenges in crop improvement, offering powerful tools for analyzing complex phenotypic data and predicting key agronomic traits. For example, Varela et al. used high-temporal resolution Unmanned Aerial Vehicle (UAV) imagery and a Random Forest approach to predict end-of-season above-ground biomass in a diverse collection of biomass sorghum, finding that dynamic traits related to early-season growth rates were highly influential predictors [40]. Beyond prediction, machine learning is also valuable for classifying and differentiating crop genotypes. Santana et al. demonstrated using UAV-acquired multispectral data and various machine learning algorithms (including Artificial Neural Networks, Random Forest, and SVM) to discriminate between six sorghum hybrids [41]. They found that spectral bands alone, particularly when used with Artificial Neural Networks, and vegetation indices, especially with Random Forest, provided accurate classification [41]. Further demonstrating the utility of machine learning for assessing sorghum biomass, Habyarimana and Baloch evaluated machine learning models for in-season biomass yield prediction in commercial sorghum fields [42].

While the aforementioned studies focused on predicting or classifying distinct genotypes or hybrids, we initially attempted a different, more granular approach: directly classifying individual sorghum accessions based on phenotypic traits. This proved challenging, yielding low accuracy. A key reason for this is the significant phenotypic overlap among many of these diverse landraces. Many of these accessions are not uniquely distinguishable based on this limited trait set, meaning the models could not identify a clear, separable pattern for each of the 179 classes. This highlights a fundamental difficulty in applying classification models to diverse germplasm collections compared to studies with more clearly defined and separated groups. With only nine phenotypic traits, it was challenging for the models to reliably distinguish between so many fine-grained classes based on the limited data available per class. To address this, we adopted a two-step approach: First, hierarchical clustering grouped accessions into four distinct clusters (Fig 3a); second, machine learning models were trained to predict these cluster assignments. We selected a diverse range of algorithms for this task, including Logistic Regression (a standard linear model), Ridge Classifier (a regularized linear model), SVM with an RBF kernel (capable of capturing non-linear relationships), Random Forest and Boosted Tree (ensemble methods known for handling complex interactions), KNN (a non-parametric instance-based method), and Decision Tree (a simple, interpretable model). This selection aimed to cover a broad spectrum of modeling approaches, from simple linear models to complex non-linear and ensemble methods, to comprehensively assess their ability to classify accessions into the pre-defined clusters.

When classifying sorghum accessions into pre-defined clusters using an undersampled dataset, several machine learning models performed well. The SVM with an RBF kernel achieved the highest validation accuracy (0.94), closely followed by Logistic Regression (0.92), Boosted Trees (0.90), KNN (0.90), and the Decision Tree (0.90) (Table 3). Although the

Ridge Classifier achieved the lowest validation accuracy (0.82), its performance was still reasonably good. This is likely because the preceding hierarchical clustering step reduced the complexity of the classification task by grouping similar accessions, making it easier for the linear Ridge Classifier to find separating hyperplanes. Performance on the independent test set generally mirrored the validation results, with Logistic Regression achieving the highest test accuracy (0.96) and the Ridge Classifier achieving the lowest (0.76). This demonstrates that even with an undersampled dataset, several models can effectively predict cluster membership based on the phenotypic traits.

After addressing class imbalance using the ADASYN oversampling technique, all tested machine-learning models achieved near-perfect performance, with accuracy, precision, recall, and F1-score all reaching 0.99. This outcome demonstrates the effectiveness of ADASYN in creating a balanced dataset for the classification task (Table 4). This near-perfect performance indicates that the cluster classification task becomes significantly easier with a balanced dataset. Also, when the synthesized data is combined with the original dataset, it generates a significant amount of additional data points, which diminishes the relevance of the specific algorithm choice. The high accuracies achieved through this oversampling strongly suggest that the four clusters identified by hierarchical clustering are, in fact, phenotypically distinct and well-separated groups. For breeding programs, this high degree of classifiability is significant, as it demonstrates that machine learning can be a powerful tool for rapidly sorting diverse germplasm into meaningful categories based on key traits, potentially streamlining the selection of parental lines for targeted improvement. First and foremost, it is crucial to acknowledge that the near-perfect classification scores are likely an artifact of the methodological approach. Applying ADASYN to the entire dataset before splitting, as was done in this analysis, can cause information leakage and lead to overestimated performance. Despite this inflation, the ease with which the models achieved near-perfect separation provides strong indirect evidence that the four clusters identified by hierarchical analysis are indeed phenotypically distinct and well-separated. Beyond this primary methodological consideration, it is also important to recognize that the synthetic data points it generates may not perfectly reflect the proper distribution of the minority classes. Future work could therefore explore alternative oversampling techniques or cost-sensitive learning methods to mitigate potential biases introduced by synthetic data.

### Predicting grain yield and identifying key influential traits

For grain yield prediction, a similar diverse set of models was chosen, encompassing linear and non-linear approaches: Neural Boosted, Bootstrap Forest, SVM-RBF, Boosted Tree, Fit Stepwise, Generalized Regression Lasso, Fit Least Squares, KNN, and Decision Tree. This range of models allowed us to explore different assumptions about the underlying relationships between traits and yield and to identify which types of models are most effective for this specific prediction task, a modern approach to the longstanding agricultural goal of evaluating sorghum varieties for yield and other key traits [38]. A key objective of this study was to train accurate machine learning models for predicting grain yield, a complex trait of paramount importance in sorghum breeding. Across a range of machine learning models evaluated using cross-validation, the Neural Boosted model, specifically the NTanH(3)NBoost(8) configuration, exhibited the best predictive performance for grain yield. This model achieved an average R-squared of 0.3597 and an average RASE of 4.873 on the validation sets (Table 5). While this average R-squared value is modest, it represents a meaningful prediction level achieved using only pre-harvest phenotypic traits, highlighting the inherent challenge and value of phenotype-only yield prediction in the absence of environmental or genotypic data. Following the Neural Boosted model, Bootstrap Forest (R-squared = 0.3327, RASE = 4.9657) and SVM-RBF (R-squared = 0.2917, RASE = 5.1306) also showed moderate predictive ability. In contrast, other models, such as the Decision Tree (R-squared = 0.1777, RASE = 5.5271), exhibited lower performance. The highest R-squared value in the Neural Boosted model is 0.4784. The consistency across validations is the best among the models, with the lowest SD RASE being 0.57 in the Boosted Tree model.

Building on this, Ferraz et al. also successfully applied neural networks (Multilayer Perceptrons) to estimate sorghum grain yield in tropical environments, achieving an R-squared of up to 0.89 and RMSE of 0.22 t/ha when combining

vegetation indices and soil elevation data [43]. This further validates the potential of neural networks for sorghum yield prediction and highlights the value of integrating multiple data sources. Our study, while achieving an average R-squared of 0.3597 for grain yield prediction using a Neural Boosted model, primarily focuses on integrating machine learning for both yield prediction and germplasm characterization using phenotypic data only, a combination less explored in previous research. Khaki and Wang, using a deep neural network on maize data from the Syngenta Crop Challenge, reported an RMSE of 12% of the average yield (and 50% of the standard deviation) when using predicted weather data, and 11% (and 46% of the standard deviation) with perfect weather data [44]. While a direct comparison of R-squared values is difficult due to differences in datasets (crops, traits, and environments) and evaluation metrics, our results demonstrate the feasibility of predicting sorghum yield with reasonable accuracy using only pre-harvest phenotypic data. In contrast, Khaki and Wang inputted both genotypic & environmental dataset [44]. A recent study employed Random Forest to forecast sorghum yield patterns throughout the U.S. It found that the most significant factors were irrigation methods, vapor pressure, and time, resulting in an anticipation of an average yield reduction of 2.7% from 2018 to 2099 due to climate change [45]. These comparisons, encompassing both Neural Network and other machine learning approaches, highlight the overall potential of these techniques for capturing complex, non-linear relationships between phenotypic traits, environmental factors, and yield.

An examination of feature importance in the leading models (Neural Boosted, Bootstrap Forest, SVM-RBF, and Boosted Tree) uncovered both common themes and some noteworthy variations. All four models consistently identified seed weight as the most significant predictor, followed closely by other characteristics such as germination rate, panicle length, and height. It is assumed that this is partially attributable to the direct relationship between individual seed weight and the three-panicle weight used to measure yield. However, it also aligns with established physiological understanding. This observation is in line with established physiological insights: a larger seed weight typically suggests higher amounts of carbohydrates, proteins, and other essential nutrients in the endosperm [46,47], which equips the developing seedling with a more robust initial energy source. Similarly, a higher germination rate contributes to the success of more seedlings, thereby enhancing yield potential. However, the models also differed in their prioritization of other traits. The Neural Boosted model placed greater emphasis on panicle height, while SVM-RBF prioritized panicle length. These variations likely reflect how these algorithms capture complex interactions, suggesting that a multi-model approach can provide a more comprehensive understanding of how traits influence yield. For instance, panicle height, although not directly contributing to grain number, may reflect overall plant vigor and photosynthetic capacity, which in turn influence the plant's ability to fill grains effectively. Panicle length, on the other hand, is more directly related to the potential number of grain-bearing spikelets. Interestingly, disease resistance traits had relatively low importance in our yield prediction models. There are several potential reasons for this observation. On a practical level, it could be due to relatively low disease pressure in the field trials, which may restrict the visibility of resistance differences, or the accessions examined may have comparatively high resistance levels, reducing yield variation caused by disease. It is also possible that the models could have inherent limitations in accurately representing the complex and possibly non-linear impacts of disease resistance on yield. However, beyond these methodological considerations, this finding may point to a more fundamental biological strategy. We hypothesize, therefore, that this reflects a strategy of seed vigor priority, where under the specific environmental pressures of these regions, securing initial seedling establishment through robust seed resources (i.e., higher seed weight and germination) represents a more critical and consistent survival strategy than investing resources in defense against a variable spectrum of pathogens. In other words, the evolutionary pressure to win the competition for early establishment may have been a more primary driver in these landraces than the pressure to combat pathogens. Further investigation, potentially employing different modeling approaches, targeted field trials, and genomic analysis, is warranted to explore these multifaceted aspects further.

The varying effectiveness of different machine learning models in clustering analysis, indicated by the silhouette scores, deserves careful attention. Comparably higher silhouette scores for the MLPRegressor in combination with

K-Means (0.317) and for the GMM applied directly to the scaled data (0.311) imply that these approaches achieved more apparent cluster separations compared to others. Conversely, Random Forest (0.115), Bagging classifier (0.127), KNN (0.087), and Decision Tree (0.122) displayed lower silhouette scores, signifying less distinct clustering. These variations underscore the sensitivity of clustering outcomes to the choice of algorithm and its underlying assumptions about the data's structure. It's important to note that even the higher silhouette scores, which indicate better separation, are still relatively modest, indicating only weak separation between the identified groups. This, coupled with the high classification accuracy after applying ADASYN, suggests that while the four clusters identified by hierarchical clustering are statistically distinguishable based on the phenotypic traits, they may not represent highly distinct, well-separated biological groups. The true underlying population structure may be more complex, possibly involving a mix of distinct subgroups and continuous variation gradients. Taken together, these results demonstrate the dual utility of the PIML framework: it not only produces a predictive model for a key agronomic trait but also reveals subtle phenotypic structures within the germplasm. The consistent identification of seed quality as a primary driver for yield, coupled with the varied performance of clustering algorithms, underscores the necessity of selecting appropriate ML tools to answer specific biological questions.

## Limitations and future directions

Several limitations of this study should be acknowledged. Our initial attempt to classify all 179 accessions individually was constrained by a challenging feature-to-sample ratio. With only nine phenotypic traits, there was insufficient data to distinguish between so many fine-grained classes, highlighting a limitation of applying such models to diverse germplasm without a larger feature set. The reliance on phenotypic data alone, while informative, does not capture the full complexity of genotype-by-environment interactions. Environmental conditions, experimental design, and measurement techniques can introduce variability and potentially bias the observed traits and, consequently, the clustering and prediction results. Furthermore, potential overfitting on the training sets for some models and the nuances of using sampling techniques to address class imbalance should be considered. While machine learning offers significant potential for accelerating crop improvement, practical implementation in breeding programs faces challenges. These include building a robust data infrastructure for managing large-scale phenotypic and genotypic data, acquiring expertise in data science and machine learning, and carefully evaluating the costs and benefits compared to traditional methods. Integrating machine learning-derived insights into breeding decisions requires close collaboration between data scientists, breeders, and agronomists.

Future research should therefore prioritize the integration of multi-modal data to overcome these limitations. Incorporating high-density genotypic markers from techniques like genotyping-by-sequencing (GBS) will be critical to enhance prediction accuracy and yield deeper biological insights into the genetic basis of key traits, enabling powerful next steps such as genome-wide association studies (GWAS). Simultaneously, integrating detailed environmental variables, such as soil characteristics and weather data, is crucial for developing more robust models that can account for genotype-by-environment (G×E) interactions. As datasets become larger and more complex, exploring alternative clustering algorithms and advanced deep learning approaches, like convolutional neural networks (CNNs) for image-based phenotyping, will also be vital for accelerating sorghum improvement efforts.

## Conclusion

Our PIML framework has clearly demonstrated that the key to increasing sorghum yield in Ethiopia and Senegal lies in the 'seed' itself. The analysis highlighted that non-linear models like the Neural Boosted configuration were most effective for prediction, while clustering algorithms such as GMM and MLP-driven K-Means best discerned the underlying germplasm structure. This underscores the importance of selecting appropriate machine learning tools for specific analytical tasks in plant breeding.

Our analysis consistently identified seed weight and germination rate as the primary determinants of grain yield potential across multiple top-performing models. In contrast, disease resistance traits were less influential in this specific dataset, suggesting that under these field conditions, inherent seed vigor was a more critical factor for yield than pathogen defense.

The high classification accuracies achieved by utilizing the ADASYN method to tackle class imbalance demonstrate the power of machine learning in distinguishing germplasm according to phenotypic characteristics, even when those traits appear very similar. It is important to note, however, that while effective, ADASYN relies on generating synthetic data, which may not perfectly capture the true distribution of the minority classes and can introduce subtle biases. The generalizability of these findings should also be considered in light of the study's specific focus on Ethiopian and Senegalese accessions and a limited set of phenotypic traits. Nevertheless, the methodologies and key findings, particularly the emphasis on seed quality traits, have broader applicability to breeding programs for other cereal crops like maize and wheat (*Triticum aestivum*) facing similar challenges in optimizing yield. This claim must be balanced by the recognition that the relative importance of specific traits may differ; for example, the direct contribution of seed weight to yield may vary in crops with different reproductive strategies and grain development patterns.

Ultimately, the insights obtained highlight the important role of integrating machine learning into plant breeding. This integration enables a more efficient and data-driven selection of superior genotypes, accelerating the development of accessions that are better adapted to specific environments and various biotic or abiotic challenges. Specifically, prioritizing seed weight and germination rate during early-generation selection, a strategy supported by our models, offers a practical path toward enhancing breeding efficiency. Based on these findings, the evidence suggests that prioritizing the use of high-quality seed with verified high germination rates and seed weight would be beneficial for farmers in the study regions, though we acknowledge that access to and the cost of such seed may present challenges in resource-limited settings. Furthermore, establishing local seed testing facilities and promoting best practices for on-farm seed storage are crucial steps that require sustained investment in local infrastructure and training to ensure seed quality and maximize yield potential. Combining the use of superior seed with appropriate agronomic practices will further enhance sorghum productivity.

## Acknowledgments

We thank the reviewers for their constructive feedback. We would also like to thank Yoonjung Lee for her contributions during the initial stages of this project. This work was supported by the U.S. Department of Agriculture, Agricultural Research Service. Mention of any trade names or commercial products in this article is solely for the purpose of providing specific information and does not imply recommendation or endorsement by the U. S. Department of Agriculture. USDA is an equal opportunity provider and employer, and all agency services are available without discrimination.

## Author contributions

**Conceptualization:** Ezekiel Ahn, Sunchung Park.

**Data curation:** Louis K. Prom.

**Formal analysis:** Ezekiel Ahn, Insuck Baek, Adama R. Tukuli, Seunghyun Lim.

**Funding acquisition:** Moon S. Kim, Lyndel W. Meinhardt.

**Methodology:** Insuck Baek, Sunchung Park, Clint Magill.

**Project administration:** Ezekiel Ahn.

**Resources:** Louis K. Prom, Lyndel W. Meinhardt, Clint Magill.

**Software:** Ezekiel Ahn, Insuck Baek, Adama R. Tukuli, Seunghyun Lim.

**Supervision:** Ezekiel Ahn, Moon S. Kim, Lyndel W. Meinhardt, Clint Magill.

**Validation:** Seok Min Hong.

**Visualization:** Jae Hee Jang.

**Writing – original draft:** Ezekiel Ahn.

**Writing – review & editing:** Louis K. Prom, Jae Hee Jang, Adama R. Tukuli, Seok Min Hong, Sunchung Park.

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
