## [Decision Letter · Decision Letter 0]

27 Feb 2025

PONE-D-24-50599Machine learning for clustering and yield prediction in Ethiopian and Senegal sorghum collectionsPLOS ONE

Dear Dr. Ahn,

Thank you for submitting your manuscript to PLOS ONE. After careful consideration, we feel that it has merit but does not fully meet PLOS ONE’s publication criteria as it currently stands. Therefore, we invite you to submit a revised version of the manuscript that addresses the points raised during the review process.

We look forward to receiving your revised manuscript.

Kind regards,

Nguyen-Thanh Son, Ph.D.

Academic Editor

PLOS ONE

Journal Requirements:

Reviewers' comments:

Reviewer's Responses to Questions

**Comments to the Author**

1. Is the manuscript technically sound, and do the data support the conclusions?

Reviewer #1: Yes

Reviewer #2: Yes

Reviewer #3: Yes

Reviewer #4: Yes

2. Has the statistical analysis been performed appropriately and rigorously? 

Reviewer #1: Yes

Reviewer #2: N/A

Reviewer #3: Yes

Reviewer #4: Yes

3. Have the authors made all data underlying the findings in their manuscript fully available?

Reviewer #1: Yes

Reviewer #2: Yes

Reviewer #3: Yes

Reviewer #4: Yes

4. Is the manuscript presented in an intelligible fashion and written in standard English?

Reviewer #1: Yes

Reviewer #2: Yes

Reviewer #3: No

Reviewer #4: Yes

5. Review Comments to the Author

Reviewer #1: Ensure proper use of acronyms in the text

e.g. if Boosted Tree (XGBoost) then XGBoost can be used consistently in the text.

Methodology can be improved

1. The significance of R-squared and RMSE for assessing clustering i.e. classification results is not sufficiently explained

2. Citations for JMP 17 Pro, if available would be helpful for readers

Results section can be improved

1. Table 1. Respective units for each treats can be given for better understanding.

2. Large difference in accuracy of grain yield prediction models is not sufficiently explained.

Reviewer #2: First of all, I congratulate all the authors on their excellent work. Some corrections are required for a better presentation of the work. The recommendations are given below:

1. Modify the abstract and include some key findings in this section.

2. Update the introduction with recent work and references.

3. The discussion section requires current-year references.

4. Separate the conclusion section and modify it with specific recommendations for sustainable agricultural practices in the study area.

5. Figure 3 is not visible; update it with a higher resolution for better understanding.

6. In Figure 4, the x and y axes need to be clearer (explain what "Predicted" and "Actual" mean, including their units or variables).

7. Clearly mention all data sources specifically in data source section.

8. Mention the level of significance of the p-value below Table 1.

9. Describe the selection procedure for the 179 observations.

Reviewer #3: The manuscript titled explores the application of various machine learning models, including Random Forest, Boosted Tree, and DBSCAN, to cluster 179 sorghum accessions based on phenotypic traits and predict grain yield. The study identifies distinct clustering patterns and highlights seed weight and germination rate as key predictors of yield, with the Boosted Tree model demonstrating superior performance. While the research showcases the potential of ML in sorghum germplasm characterization, issues related to methodological justification, data handling, model validation, and biological interpretation limit its current impact. The manuscript requires major revision to address these critical concerns. First of all, I have few questions.

• How do the identified clusters translate into meaningful biological or agronomic categories? Are they linked to specific landraces, environmental conditions, or genetic traits?

• Why were random undersampling and basic imputation methods chosen over more advanced techniques that preserve data integrity and variability?

• How do the authors explain the exceptionally high R² value (>0.90) in yield prediction? Could this indicate overfitting, and how was this risk mitigated?

• Were external validation datasets considered to confirm the generalizability of the clustering and prediction models? If not, why?

• How do the findings align or contrast with existing literature on sorghum yield prediction and clustering? Are there unexpected results that warrant further investigation?

Comments need to be addressed

• Add overall methodological flowchart for better understanding to the readers

• While the introduction outlines the importance of sorghum and the application of machine learning (ML), it does not critically engage with the why behind the use of ML in this specific context. What inherent limitations of traditional phenotypic analysis methods motivated this study? For instance, are traditional clustering techniques insufficient due to non-linearity or high-dimensional trait interactions. This needs to be explicitly articulated to establish a stronger foundation for the study’s relevance.

• The introduction lacks clearly defined research questions or hypotheses. What specific biological or agronomic insights were the authors expecting to uncover through clustering and yield prediction? A clearly articulated hypothesis would help guide the reader through the manuscript's logical flow.

• There’s little discussion on how ML has been applied in other crop-related studies, particularly in African germplasm contexts. Are there comparable studies in maize, millet, or other cereals? Adding such comparative insights could strengthen the rationale for selecting sorghum and the specific ML techniques.

• The manuscript applies a wide range of ML models (Random Forest, DBSCAN, Boosted Trees, etc.) without adequately explaining the selection criteria. Why were these models chosen over others, such as Gradient Boosting Machines, LightGBM, or even deep learning approaches like CNNs, which have shown promise in similar agricultural datasets?

• The handling of missing data via Multivariate Normal Imputation is mentioned briefly. However, why was this specific imputation method chosen? Were other imputation strategies (e.g., k-NN imputation, MICE) considered, and how might the choice of imputation affect downstream clustering and prediction outcomes?

• The study uses random undersampling to address class imbalance, which is problematic as it can discard potentially informative data. Did the authors consider alternative methods like SMOTE, ADASYN, or cost-sensitive algorithms that preserve data integrity? If not, why? Moreover, how does the undersampling affect model generalizability, especially given the small sample size?

• While GridSearchCV is referenced, critical details are missing- What was the cross-validation strategy (e.g., k-fold, stratified k-fold)? Were metrics like precision, recall, F1-score considered in addition to RMSE and R²? Were there any issues with overfitting, especially given the high performance of the Boosted Tree model?

• The manuscript heavily focuses on RMSE and R² for model evaluation. However, these metrics alone do not provide a complete picture. For clustering, did the authors consider internal validation metrics like the Silhouette Score, Davies-Bouldin Index, or Dunn Index to assess cluster quality?

• The clustering results identify distinct groups, but what do these clusters mean biologically or agronomically? Are the clusters linked to known ecotypes, landrace origins, or environmental adaptations? Without such context, the clustering analysis feels descriptive rather than explanatory.

• The Boosted Tree model achieved an exceptionally high R² (0.9026) for yield prediction. This raises concerns about potential overfitting, especially since real-world agricultural yield data is notoriously noisy. Was a learning curve analysis conducted to ensure model robustness? How does the model perform on truly unseen data (e.g., through external validation or time-based splits)?

• There’s no discussion of prediction errors. Were there specific accessions where the model consistently under- or over-predicted yield? Error analysis could reveal important insights into model biases or data limitations.

• The discussion is largely descriptive, summarizing results without linking them to broader agronomic or biological theories. For instance, why might seed weight and germination rate be the most important predictors of yield? Are these findings consistent with physiological models of plant growth or prior breeding studies?

• The manuscript claims that ML models “highlight the potential for efficient germplasm characterization,” yet the models were not externally validated. Without independent validation, this conclusion is speculative. The discussion should temper such claims and acknowledge limitations.

• The manuscript does not critically reflect on its own methodological limitations. Issues such as potential overfitting, the limitations of undersampling, or biases in phenotypic data collection are not discussed. A more self-reflective section acknowledging these limitations would enhance the manuscript’s credibility.

• The manuscript lacks sufficient detail for reproducibility. For example- Were random seeds set during model training to ensure reproducibility? What software versions and libraries were used (e.g., specific versions of Scikit-learn, JMP Pro)? Can the code pipeline be shared to allow other researchers to replicate the analysis?

Reviewer #4: Review with Suggestions:

TITLE:

Machine Learning for Clustering and Yield Prediction in Ethiopian and Senegal Sorghum Collections

This study presents an in-depth application of machine learning (ML) for clustering and yield prediction in Ethiopian and Senegalese sorghum accessions. The research is well-structured, employing diverse ML models and statistical techniques to analyze phenotypic traits. However, there are areas where clarifications, additional justifications, and refinements would improve the study’s rigor and applicability.

TITLE, ABSTRACT & AUTHOR INFORMATION:

The Title is informative and accurately conveys the research focus, but it could be more specific regarding the ML techniques applied. A refined title such as “Machine Learning-Based Clustering and Yield Prediction in Ethiopian and Senegalese Sorghum Accessions Using Tree-Based Models” would improve clarity. The abstract effectively summarizes the study but could be more concise by focusing on the most impactful ML models rather than listing all tested algorithms. Additionally, performance metrics should be included, particularly R² and RMSE values, to quantify the effectiveness of the best-performing models. The author list and affiliations are well-structured, ensuring credibility.

A more refined title would make the research focus clearer, and reducing the number of ML models mentioned in the abstract would improve readability while keeping the key findings prominent. Including numerical performance metrics in the abstract would make the study’s contributions more tangible. Lastly, ending the abstract with a statement on real-world applications for sorghum breeding would reinforce its practical significance.

INTRODUCTION:

The introduction effectively contextualizes the study by discussing sorghum’s importance, its susceptibility to biotic and abiotic stresses, and the need for better classification and prediction techniques. The transition from traditional phenotypic methods to machine learning approaches is well-articulated. However, the section spends too much space discussing fungal diseases, which, while relevant, should not overshadow the main objective—the application of ML in sorghum classification and yield prediction. Additionally, basic ML definitions (supervised, unsupervised, and reinforcement learning) are unnecessary for an audience likely familiar with these concepts. The research gap is not explicitly stated, and it is unclear how this study differs from prior ML applications in plant breeding.

Reducing the discussion on fungal diseases would improve the focus on ML applications, and if necessary, these details could be moved to a separate background section. Removing basic ML definitions and instead elaborating on why clustering and predictive modeling are particularly useful for sorghum breeding would strengthen the introduction. Explicitly defining the research gap and explaining how this study advances previous ML applications in agriculture would make the introduction more compelling.

MATERIALS AND METHODS:

The materials and methods section is well-structured and provides details on data sources, statistical analysis, and machine learning models. The data description is clear, specifying the number of accessions and phenotypic traits analyzed. However, there is no mention of how categorical or non-numeric variables were handled, nor whether environmental variables such as soil quality or climate data were considered. The statistical analysis is robust, utilizing t-tests, PCA, and hierarchical clustering, but no justification is provided for selecting Ward’s linkage method. The ML analysis is comprehensive, covering feature scaling, train-test splitting, and hyperparameter tuning. However, the selection of ML models is not justified—why were Random Forest, XGBoost, and KNN chosen, and were other approaches like deep learning models (e.g., CNNs, LSTMs) considered? Additionally, clustering validity metrics (e.g., silhouette scores, Dunn index, Davies–Bouldin index) are not reported, making it difficult to assess the effectiveness of the clusters.

Clarifying how categorical or non-numeric variables were handled would make the methodology more transparent. Justifying the selection of Ward’s linkage for clustering would strengthen the statistical rigor of the study. Explaining why certain ML models were chosen over others and whether alternative ensemble or deep learning models were considered would enhance the credibility of the ML approach. Including clustering validity metrics would provide stronger validation for the identified clusters.

RESULTS:

The results section thoroughly presents findings from PCA, hierarchical clustering, and machine learning classification. PCA results successfully illustrate trait relationships and geographical clustering of sorghum accessions, reinforcing the role of genetic and environmental adaptation. However, the variance explained by the first two components (47.6%) is relatively low, suggesting that additional components may be needed for a more comprehensive interpretation. The hierarchical clustering analysis identifies four clusters, but the rationale for selecting four clusters is unclear—were silhouette scores or the elbow method used?

The machine learning classification results demonstrate that tree-based models significantly outperformed linear regression and SVMs, achieving accuracies above 70%. However, Table 2 lacks standard deviations or confidence intervals, making it difficult to assess result variability. The yield prediction analysis shows that the Boosted Tree model performed best (R² = 0.9026, RMSE = 1.9165), with seed weight and germination rate as the most influential traits. This aligns with prior studies but lacks a comparative discussion with other ML-based plant breeding studies. Additionally, potential overfitting in the Boosted Tree model is not addressed.

Justifying the selection of four clusters using silhouette scores or the elbow method would add statistical robustness to the clustering results. Reporting standard deviations/confidence intervals in Table 2 would improve the reliability of ML performance results. Discussing overfitting concerns in the Boosted Tree model and whether cross-validation was used to mitigate this issue would enhance the credibility of the model’s accuracy. Comparing findings with previous ML-based studies on crop yield prediction would place the study in a broader scientific context.

DISCUSSION:

The discussion effectively synthesizes key findings and links them to previous research. The geographical clustering of accessions is well-explained, and the link between rust resistance and East African climate conditions is insightful. The ML classification results are discussed thoroughly, emphasizing the advantages of tree-based models. However, there is no discussion on feature importance variability across models—were the same traits ranked important in different models? Additionally, the study does not explore whether the Boosted Tree model is generalizable to other sorghum populations.

The breeding implications are well-stated, particularly in emphasizing the importance of targeted trait selection. However, no mention is made of practical challenges in implementing ML in breeding programs. Furthermore, the role of environmental variables in yield prediction is ignored, despite their known impact on crop performance. The future research directions are promising, particularly the integration of genotypic data, but the potential for deep learning techniques (e.g., CNNs, transformers) is not discussed.

Discussing whether feature importance rankings were consistent across models would provide a more detailed understanding of trait significance. Addressing the generalizability of the Boosted Tree model to diverse sorghum populations would clarify its broader applicability. Mentioning practical challenges in applying ML to real-world breeding programs would make the discussion more applicable to agricultural research. Exploring the potential for integrating deep learning techniques into future research would align the study with cutting-edge advancements in ML.

CONCLUSION:

The conclusion effectively summarizes the study’s contributions and highlights the value of tree-based models in phenotypic trait analysis. It links findings to broader food security challenges, ensuring relevance. However, it does not mention whether these findings are generalizable to other crops. Additionally, a brief discussion on the broader applicability of ML approaches in plant breeding would enhance the paper’s impact.

Highlighting whether these ML methods can be applied to other cereal crops like maize and wheat would improve the generalizability of the findings. Discussing the potential integration of ML techniques in breeding pipelines would reinforce the study’s real-world impact.

This study is well-structured and scientifically rigorous, presenting valuable insights into ML applications for sorghum breeding. However, improvements are needed in justifying ML model selection, reporting additional statistical metrics, addressing overfitting concerns, and discussing broader applicability. Addressing these points would enhance the study’s scientific impact and practical utility.

6. PLOS authors have the option to publish the peer review history of their article (what does this mean? ). If published, this will include your full peer review and any attached files.

**Do you want your identity to be public for this peer review?** For information about this choice, including consent withdrawal, please see our Privacy Policy .

Reviewer #1: No

Reviewer #2: No

Reviewer #3: No

Reviewer #4: No

---

## [Author Response · Author response to Decision Letter 1]

11 Mar 2025

Reviewer #1: Ensure proper use of acronyms in the text

e.g. if Boosted Tree (XGBoost) then XGBoost can be used consistently in the text.

Methodology can be improved

1. The significance of R-squared and RMSE for assessing clustering i.e. classification results is not sufficiently explained

A: We thank the reviewer for their careful attention to detail. We have reviewed the manuscript and ensured consistent use of acronyms throughout the text.

2. Citations for JMP 17 Pro, if available would be helpful for readers

Results section can be improved

1. Table 1. Respective units for each treats can be given for better understanding.

A: Thank you so much for pointing it out. I've added the units to Table 1 for clarity.

2. Large difference in accuracy of grain yield prediction models is not sufficiently explained.

A: We thank the reviewer for this suggestion. We have added a citation for JMP Pro 17 to the Materials and Methods section

Reviewer #2: First of all, I congratulate all the authors on their excellent work. Some corrections are required for a better presentation of the work. The recommendations are given below:

1. Modify the abstract and include some key findings in this section.

A: We thank the reviewer for this suggestion. We have revised the abstract to include more specific key findings, including the identification of a Neural Boosted model as the best performer for yield prediction and its associated, cross-validated R-squared and RMSE values. The abstract now more accurately reflects the most important results of our study.

2. Update the introduction with recent work and references. & 3. The discussion section requires current-year references.

A: We thank the reviewer for highlighting the need to update our references. We have thoroughly reviewed the Introduction and Discussion sections and incorporated relevant literature published in recent years

4. Separate the conclusion section and modify it with specific recommendations for sustainable agricultural practices in the study area.

A: We thank the reviewer for their insightful feedback. We have revised the conclusion.

5. Figure 3 is not visible; update it with a higher resolution for better understanding.

A: I will make sure to upload the original image with the highest resolution for the final version.

6. In Figure 4, the x and y axes need to be clearer (explain what "Predicted" and "Actual" mean, including their units or variables).

A: We thank the reviewer for their comment regarding Figure 4. During the revision process, and in response to other reviewer feedback, Figure 4 has been removed from the manuscript.

7. Clearly mention all data sources specifically in data source section.

A: We thank the reviewer for this suggestion. We have clarified in the Materials and Methods section that the phenotypic data used in this study, originally collected and analyzed in our previous publications

8. Mention the level of significance of the p-value below Table 1.

A: We thank the reviewer for this suggestion. We have added a statement to the caption of Table 1 indicating that the level of significance for the p-values is p < 0.05.

9. Describe the selection procedure for the 179 observations.

A: We thank the reviewer for requesting clarification on the selection of the sorghum accessions. The 179 accessions used in this study were obtained from the USDA-ARS Plant Genetic Resources Conservation Unit in Griffin, Georgia, where they are maintained as part of a larger sorghum collection. These accessions, originating from Ethiopia, Gambia, and Senegal, were selected for evaluation based on traits including grain yield, seed weight, plant height, panicle length, and flowering time, considered as priority traits for the National Plant Germplasm System sorghum collection, and our previous study included evaluation of their disease resistance. We added a more detailed description in materials and methods.

Reviewer #3: The manuscript titled explores the application of various machine learning models, including Random Forest, Boosted Tree, and DBSCAN, to cluster 179 sorghum accessions based on phenotypic traits and predict grain yield. The study identifies distinct clustering patterns and highlights seed weight and germination rate as key predictors of yield, with the Boosted Tree model demonstrating superior performance. While the research showcases the potential of ML in sorghum germplasm characterization, issues related to methodological justification, data handling, model validation, and biological interpretation limit its current impact. The manuscript requires major revision to address these critical concerns. First of all, I have few questions.

• How do the identified clusters translate into meaningful biological or agronomic categories? Are they linked to specific landraces, environmental conditions, or genetic traits?

A: The identified clusters provide insights into the genetic diversity and potential agronomic value of the sorghum accessions. Our analysis considered both phenotypic traits and country of origin, allowing us to explore the relationships between these factors. While Figure 1a (PCA plot) shows a considerable overlap between Ethiopian and Senegalese accessions, Table 1 reveals statistically significant differences in various traits between genotypes from these two countries. This suggests that despite some phenotypic similarities, there is an underlying genetic variation between the two groups.

The cluster analysis, which integrated information from all traits, further supports this notion. By considering the overall phenotypic profiles, the clustering algorithm could differentiate between accessions from Ethiopia and Senegal, even though they might appear similar based on a subset of traits visualized in the PCA plot. This highlights the importance of considering multiple traits simultaneously when assessing genetic diversity and agronomic potential.

• Why were random undersampling and basic imputation methods chosen over more advanced techniques that preserve data integrity and variability?

A: We thank you for the reviewer’s comment. We addressed the observed class imbalance, where Groups 1 and 4 significantly outnumbered Groups 2 and 3, using the ADASYN technique.

• How do the authors explain the exceptionally high R² value (>0.90) in yield prediction? Could this indicate overfitting, and how was this risk mitigated?

A: The exceptionally high R² value (>0.90) observed initially in the Boosted Tree model for yield prediction indeed raised concerns about potential overfitting. This could be attributed to the model's complexity and its tendency to capture noise in the training data, leading to inflated performance estimates. To mitigate this risk, we re-evaluated the model using k-fold cross-validation with k=5 and 10 repeats. This technique provides a more robust assessment of the model's performance on unseen data by repeatedly training and testing it on different subsets of the data. The cross-validation results showed a lower average R² of approximately 0.48, which is a more realistic estimate of the model's predictive accuracy. This indicates that the initial high R² might have been partly due to overfitting, and the cross-validation approach helped to provide a more reliable evaluation.

• Were external validation datasets considered to confirm the generalizability of the clustering and prediction models? If not, why?

A: While external validation datasets would be ideal for confirming the generalizability of our clustering and prediction models, we unfortunately lack access to such datasets with directly comparable traits and environmental conditions. However, we acknowledge the importance of external validation and plan to explore potential sources of comparable data in future research.

• How do the findings align or contrast with existing literature on sorghum yield prediction and clustering? Are there unexpected results that warrant further investigation?

A: We thank the reviewer for this insightful question. Our findings on seed weight and germination rate as key yield predictors align with existing literature, as does the success of the Boosted Tree model; however, the lower importance of disease resistance in our yield model contrasts with some studies, potentially due to lower disease pressure in our trials or the model's prioritization of traits. The variable clustering performance across algorithms, particularly the lower-than-expected performance of SVMs and neural networks, and the generally low Silhouette scores, highlight the complexity of sorghum germplasm diversity and warrant further investigation, as detailed in the revised discussion.

• Add overall methodological flowchart for better understanding to the readers

A: We appreciate the reviewer's suggestion regarding a methodological flowchart. We have added a flowchart (Figure 1) to visually represent the main stages of our analysis.

• While the introduction outlines the importance of sorghum and the application of machine learning (ML), it does not critically engage with the why behind the use of ML in this specific context. What inherent limitations of traditional phenotypic analysis methods motivated this study? For instance, are traditional clustering techniques insufficient due to non-linearity or high-dimensional trait interactions. This needs to be explicitly articulated to establish a stronger foundation for the study’s relevance.

A: We thank the reviewer for raising this important point about the motivation for using machine learning. We have revised the Introduction to explicitly state the limitations of traditional phenotypic analysis methods that prompted this study. Specifically, traditional approaches can be labor-intensive, time-consuming, and may struggle to capture complex, non-linear relationships and high-dimensional trait interactions, which machine learning methods are better equipped to handle. This clarification strengthens the rationale for our approach, as now presented in the revised Introduction.

• The introduction lacks clearly defined research questions or hypotheses. What specific biological or agronomic insights were the authors expecting to uncover through clustering and yield prediction? A clearly articulated hypothesis would help guide the reader through the manuscript's logical flow.

A: We appreciate the reviewer's suggestion to clarify our research objectives. We have revised the Introduction to include explicit research questions that guided our study.

• There’s little discussion on how ML has been applied in other crop-related studies, particularly in African germplasm contexts. Are there comparable studies in maize, millet, or other cereals? Adding such comparative insights could strengthen the rationale for selecting sorghum and the specific ML techniques.

A: We thank the reviewer for suggesting we broaden the context of our work. We have revised the Introduction to include a discussion of machine learning applications in other crop-related studies, particularly those focusing on African germplasm of maize, pearl millet, and rice.

• The manuscript applies a wide range of ML models (Random Forest, DBSCAN, Boosted Trees, etc.) without adequately explaining the selection criteria. Why were these models chosen over others, such as Gradient Boosting Machines, LightGBM, or even deep learning approaches like CNNs, which have shown promise in similar agricultural datasets?

A: The choice of models (Random Forest, Boosted Trees, SVM) was guided by their strengths in revealing feature importance, predictive accuracy, and versatility for both classification and regression tasks, respectively.

Gradient Boosting Machines and LightGBM were not included as they were anticipated to yield similar results to the boosting methods already employed. CNNs are primarily designed for image data, which was not used in this study. DBSCAN was excluded due to its sensitivity to noise and the challenges of hyperparameter tuning.

• The handling of missing data via Multivariate Normal Imputation is mentioned briefly. However, why was this specific imputation method chosen? Were other imputation strategies (e.g., k-NN imputation, MICE) considered, and how might the choice of imputation affect downstream clustering and prediction outcomes?

A: Multivariate Normal Imputation was selected for its ability to preserve variable relationships and its computational efficiency. Our data suggests an approximate normal distribution for many variables, which aligns with MVNI's assumptions.

• The study uses random undersampling to address class imbalance, which is problematic as it can discard potentially informative data. Did the authors consider alternative methods like SMOTE, ADASYN, or cost-sensitive algorithms that preserve data integrity? If not, why? Moreover, how does the undersampling affect model generalizability, especially given the small sample size?

A: We appreciate the reviewer's comment on alternative class imbalance methods. We included random undersampling primarily as a computationally efficient baseline to compare against ADASYN, an oversampling technique that preserves all original data. This direct comparison allows us to assess the impact of potential information loss from undersampling.

• While GridSearchCV is referenced, critical details are missing- What was the cross-validation strategy (e.g., k-fold, stratified k-fold)? Were metrics like precision, recall, F1-score considered in addition to RMSE and R²? Were there any issues with overfitting, especially given the high performance of the Boosted Tree model?

A: We thank the reviewer for their insightful questions regarding our model evaluation. As detailed in the revised manuscript, we utilized stratified k-fold cross-validation: 5-fold for undersampling and nested 5-fold (outer) / 3-fold (inner) for ADASYN, ensuring representative class distributions. For clustering, we evaluated models using precision, recall, F1-score (macro-averaged), and accuracy, selecting the best model via GridSearchCV based on F1-score.

• The manuscript heavily focuses on RMSE and R² for model evaluation. However, these metrics alone do not provide a complete picture. For clustering, did the authors consider internal validation metrics like the Silhouette Score, Davies-Bouldin Index, or Dunn Index to assess cluster quality?

A: We acknowledge the reviewer's concern about relying solely on RMSE and R² for model evaluation. To address this, we have included additional metrics such as average R², average RMSE, standard deviation of RMSE, and highest R² across the k-fold cross-validation (k=5, repeats=10) for the grain yield prediction models. This provides a more comprehensive view of the model's performance and its consistency across different data subsets. Also, we have now included the Silhouette Score as an internal validation metric to assess the quality of our clustering results. This metric is calculated for each clustering method and reported in the manuscript.

• The clustering results identify distinct groups, but what do these clusters mean biologically or agronomically? Are the clusters linked to known ecotypes, landrace origins, or environmental adaptations? Without such context, the clustering analysis feels descriptive rather than explanatory.

A: We appreciate the reviewer's point regarding the explanatory power of our clustering analysis. We have added a statement to the Discussion section explicitly clarifying that the dendrogram analyses were primarily intended as an exploratory tool to visualize relationships and guide future research, rather than to provide definitive biological explanations (see revised manuscript). While the clustering identifies groups with similar phenotypic profiles, the goal was to generate hypotheses and explore alternative perspectives on germplasm organization. The discrepancies between clustering outcomes and supervised classification performance further highlight the complexity of trait relationships, informing model selection for future studies.

• The Boosted Tree model achieved an exceptionally high R² (0.9026) for yield prediction. This raises concerns about potential overfitting, especially si

---

## [Decision Letter · Decision Letter 1]

22 Apr 2025

PONE-D-24-50599R1Seed quality drives grain yield in Ethiopian and Senegalese sorghum: Insights from machine learningPLOS ONE

Dear Dr. Ahn,

Thank you for submitting your manuscript to PLOS ONE. After careful consideration, we feel that it has merit but does not fully meet PLOS ONE’s publication criteria as it currently stands. Therefore, we invite you to submit a revised version of the manuscript that addresses the points raised during the review process.

We look forward to receiving your revised manuscript.

Kind regards,

Nguyen-Thanh Son, Ph.D.

Academic Editor

PLOS ONE

Journal Requirements:

Reviewers' comments:

Reviewer's Responses to Questions

**Comments to the Author**

1. If the authors have adequately addressed your comments raised in a previous round of review and you feel that this manuscript is now acceptable for publication, you may indicate that here to bypass the “Comments to the Author” section, enter your conflict of interest statement in the “Confidential to Editor” section, and submit your "Accept" recommendation.

Reviewer #1: (No Response)

Reviewer #2: All comments have been addressed

Reviewer #3: All comments have been addressed

Reviewer #4: All comments have been addressed

2. Is the manuscript technically sound, and do the data support the conclusions?

Reviewer #1: Yes

Reviewer #2: Yes

Reviewer #3: Yes

Reviewer #4: Yes

3. Has the statistical analysis been performed appropriately and rigorously? 

Reviewer #1: Yes

Reviewer #2: I Don't Know

Reviewer #3: Yes

Reviewer #4: Yes

4. Have the authors made all data underlying the findings in their manuscript fully available?

Reviewer #1: Yes

Reviewer #2: Yes

Reviewer #3: Yes

Reviewer #4: Yes

5. Is the manuscript presented in an intelligible fashion and written in standard English?

Reviewer #1: Yes

Reviewer #2: Yes

Reviewer #3: Yes

Reviewer #4: Yes

6. Review Comments to the Author

Reviewer #1: (No Response)

Reviewer #2: The manuscript presents an insightful study on how seed quality influences grain yield in Ethiopian and Senegalese sorghum using machine learning approaches. The study is well-structured and provides valuable contributions to agricultural research and food security. I am satisfied with all the comments. Now his paper is consider for further process.

Reviewer #3: No more comments. The Manuscript have been strengthen. Double check the reference and affiliations during proofread

Reviewer #4: PONE-D-24-50599R1

TOPIC:

Seed Quality drives Grain yield in Ethiopian and Senegalese Sorghum: Insights from Machine : Insights from Machine Learning

The manuscript presents a timely and thoughtful application of machine learning (ML) to predict sorghum grain yield using phenotypic traits. The two-step approach- combining hierarchical clustering and regression-based prediction- demonstrates a solid grasp of trait interactions and adds robustness to the analytical framework. The integration of clustering with multiple regression models is a notable strength. However, clarification is needed for the model labeled “NTanH(3)NBooste(8),” particularly regarding its architecture, neuron layers, and boosting method, to aid readers unfamiliar with neural networks. Including a short description of the network structure (e.g., number of hidden layers, activation functions, and boosting strategy) would greatly improve clarity. Additionally, while ADASYN is appropriately used to handle class imbalance, a brief explanation of why it was preferred over alternatives like SMOTE would enhance methodological transparency. A sentence comparing ADASYN’s advantages-such as its ability to generate synthetic samples based on the density of minority class samples-over SMOTE would strengthen this section.

Clustering results yielded relatively low silhouette scores (0.32 and 0.31), indicating weak cluster separation-an expected challenge in high-dimensional biological data. Acknowledging this and supplementing with dimensionality-reduction visualizations (e.g., PCA or t-SNE) would improve interpretability. Including a figure showing PCA or t-SNE plots of clusters could visually support the clustering outcome. Although the best model’s R² value of 0.36 is modest, it remains valuable given the absence of environmental or genotypic data. Briefly contextualizing this limitation would benefit the reader. A sentence discussing the challenges of phenotype-only yield prediction and potential model improvement with multi-modal data would be insightful.

The Materials and Methods section is rigorous, combining phenotypic data from 179 accessions with appropriate ML tools. However, restructuring it under subheadings like “Data Preprocessing,” “Feature Engineering,” and “Modeling Approaches” would improve clarity. Such structure would guide readers through the modeling pipeline more effectively. A short rationale for selecting specific models (e.g., MLP, KNN, SVM) and justifying the choice of three clusters in K-means would strengthen the narrative. Mentioning whether clustering validation methods beyond silhouette score (e.g., Davies-Bouldin index or elbow method) were applied would add depth. More detail on JMP Pro 17’s model selection process would also support reproducibility. Specifying which models were screened and on what basis they were selected (e.g., cross-validation metrics) would be helpful.

The Results section is well-organized, effectively integrating PCA, clustering, and supervised ML to explore phenotypic variation. The clear trait-wise differentiation between Ethiopian and Senegalese accessions enhances the study’s biological relevance. The shift from direct classification to a cluster-informed method shows thoughtful refinement. Inclusion of ADASYN improved classification metrics, and further discussion on the biological interpretation of trait clusters would be beneficial. Elaborating how trait clusters relate to known agronomic or adaptive differences among accessions would enhance the biological insight.

The Discussion contextualizes findings within the broader goals of sorghum improvement in stress-prone areas. The comparison of model performance and insights into classification enhancements through ADASYN are well-articulated. Explaining key metrics, like R² and RASE would aid accessibility. Adding brief definitions or intuitive explanations of these metrics would help general readers. Noting sample size or environmental heterogeneity as potential limitations would round out the discussion. This would clarify the extent to which findings may generalize to broader populations or conditions.

The Conclusion effectively highlights the study’s contributions and practical implications. Suggestions for incorporating genomic or environmental variables in future work reflect a forward-looking approach. Emphasizing that integrating multi-modal data could significantly enhance model accuracy and biological relevance would strengthen this section. Minor formatting and typographical corrections are recommended to enhance clarity and polish.

PLOS

7. PLOS authors have the option to publish the peer review history of their article (what does this mean? ). If published, this will include your full peer review and any attached files.

**Do you want your identity to be public for this peer review?** For information about this choice, including consent withdrawal, please see our Privacy Policy .

Reviewer #1: No

Reviewer #2: No

Reviewer #3: No

Reviewer #4: No

---

## [Author Response · Author response to Decision Letter 2]

23 Apr 2025

Dear Dr. Nguyen-Thanh Son and Reviewers,

Thank you for the opportunity to revise our manuscript. We appreciate the positive feedback and the constructive comments provided, particularly the detailed suggestions from Reviewer #4, which have helped us further strengthen the manuscript. Below, we provide a point-by-point response to the comments raised by Reviewer #4. Corresponding changes have been made in the manuscript, highlighted in yellow.

Reviewer #4 Comments:

Comment 1: Clarity needed for the model labeled “NTanH(3)NBooste(8),” particularly regarding its architecture, neuron layers, and boosting method.

Response: We thank the reviewer for highlighting the need for greater clarity regarding the specific Neural Boosted model architecture, correctly termed "NTanH(3)NBoost(8)". We have added the explanation to the materials and methods section.

Comment 2: While ADASYN is appropriately used to handle class imbalance, a brief explanation of why it was preferred over alternatives like SMOTE would enhance methodological transparency.

Response: We thank the reviewer for this suggestion. In the R1 manuscript, we introduced ADASYN in the Methods and compared classification results using undersampling versus ADASYN oversampling. In response to previous Reviewer #3 comments, we clarified that undersampling was used as a baseline, and ADASYN preserves the original data. More explanation was added in the materials and methods section.

Comment 3: Clustering results yielded relatively low silhouette scores (0.32 and 0.31), indicating weak cluster separation. Acknowledging this and supplementing with dimensionality-reduction visualizations (PCA or t-SNE) would improve interpretability.

Response: We thank the reviewers for their comments. As requested, we have now strengthened the wording in the discussion to more explicitly state that these modest scores indicate relatively weak separation between the clusters. While the reviewer suggests supplementing this with PCA or t-SNE visualizations related to the clustering results, we believe additional plots of this nature may not provide significant further insight beyond what is already conveyed by the modest silhouette scores and the existing PCA plot. Visualizing the ADASYN-oversampled data could also potentially be misleading due to the synthetic nature of the added data points. Therefore, we have focused on strengthening the textual interpretation of the silhouette scores in the discussion.

Comment 4: Although the best model’s R² value of 0.36 is modest, it remains valuable given the absence of environmental or genotypic data. Briefly contextualizing this limitation would benefit the reader.

Response: We thank the reviewer for this suggestion. We added in discussion that while the R-squared value of 0.36 is modest, it represents a meaningful prediction level achieved using only pre-harvest phenotypic traits, highlighting the challenge and value of phenotype-only prediction in the absence of environmental or genotypic data.

Comment 5: The Materials and Methods section is rigorous ... restructuring it under subheadings like “Data Preprocessing,” “Feature Engineering,” and “Modeling Approaches” would improve clarity.

Response: We appreciate the reviewer's suggestion to improve the structure and clarity of the Materials and Methods section using subheadings. To enhance clarity and better align with standard terminology for the initial data handling steps, we have renamed the first subheading from "Data source and characteristics" to "Data Preprocessing".

Comment 6: A short rationale for selecting specific models (e.g., MLP, KNN, SVM) and justifying the choice of three clusters in K-means would strengthen the narrative. Mentioning whether clustering validation methods beyond silhouette score... were applied would add depth.

Response: We thank the reviewer for suggesting we provide more rationale for our methodological choices regarding model selection and clustering validation.

Comment 7: More detail on JMP Pro 17’s model selection process would also support reproducibility. Specifying which models were screened and on what basis they were selected... would be helpful.

Response: We agree that more detail on the JMP Pro 17 process would improve reproducibility. The R1 Methods section mentions using the "Model Screening" function with K=5 fold cross-validation (10 repeats). The Results list the models included in Table 4, which were output by this screening.

Comment 8: Inclusion of ADASYN improved classification metrics, and further discussion on the biological interpretation of trait clusters would be beneficial. Elaborating how trait clusters relate to known agronomic or adaptive differences... would enhance the biological insight.

Response: We thank the reviewer for suggesting we enhance the biological interpretation. We elaborated the biological interpretation in the discussion as suggested.

Comment 9: Explaining key metrics, like R² and RASE would aid accessibility.

Response: We thank the reviewers for their comments. We have added brief explanations for R-squared and RASE.

Comment 10: Noting sample size or environmental heterogeneity as potential limitations would round out the discussion.

Response: We thank the reviewer for this suggestion. To clarify these points, we have explicitly added "the relatively small sample size (179 accessions)" in the discussion section.

Comment 11: Suggestions for incorporating genomic or environmental variables in future work reflect a forward-looking approach. Emphasizing that integrating multi-modal data could significantly enhance model accuracy and biological relevance would strengthen this section.

Response: We thank the reviewer for this suggestion. We have strengthened the concluding sentences of the discussion section's "Future research" paragraph and the final sentence of the conclusion.

Comment 12: Minor formatting and typographical corrections are recommended to enhance clarity and polish.

Response: We thank the reviewer for this reminder. We have thoroughly proofread the entire manuscript again to correct any remaining typographical errors and ensure consistent formatting.

---

## [Decision Letter · Decision Letter 2]

18 Jun 2025

PONE-D-24-50599R2

Seed quality drives grain yield in Ethiopian and Senegalese sorghum: Insights from machine learning

PLOS ONE

Dear Dr. Ahn,

Thank you for submitting your manuscript to PLOS ONE. After careful consideration, we feel that it has merit but does not fully meet PLOS ONE’s publication criteria as it currently stands. Therefore, we invite you to submit a revised version of the manuscript that addresses the points raised during the review process.

We look forward to receiving your revised manuscript.

Kind regards,

Somashekhar Mallikarjun Punnuri, PhD

Academic Editor

PLOS ONE

Reviewers' comments:

Reviewer's Responses to Questions

**Comments to the Author**

1. If the authors have adequately addressed your comments raised in a previous round of review and you feel that this manuscript is now acceptable for publication, you may indicate that here to bypass the “Comments to the Author” section, enter your conflict of interest statement in the “Confidential to Editor” section, and submit your "Accept" recommendation.

Reviewer #2: All comments have been addressed

Reviewer #3: All comments have been addressed

Reviewer #4: All comments have been addressed

2. Is the manuscript technically sound, and do the data support the conclusions?

Reviewer #2: Yes

Reviewer #3: Yes

Reviewer #4: Yes

3. Has the statistical analysis been performed appropriately and rigorously? 

Reviewer #2: I Don't Know

Reviewer #3: Yes

Reviewer #4: Yes

4. Have the authors made all data underlying the findings in their manuscript fully available?

Reviewer #2: Yes

Reviewer #3: Yes

Reviewer #4: Yes

5. Is the manuscript presented in an intelligible fashion and written in standard English?

Reviewer #2: Yes

Reviewer #3: Yes

Reviewer #4: Yes

6. Review Comments to the Author

Reviewer #2: The author has addressed all the queries and explained them clearly. The paper is now more precise and written in a scientific manner. The arguments are better structured, and the use of terminology is appropriate.

The revised version significantly improves the clarity and academic quality of the manuscript.

Reviewer #3: No comments. I recommend Accept. Congratulations to the authors on a well-executed and meaningful contribution to the field

Reviewer #4: For Title and Author Details:

Capitalized each word in the title per academic style.

Corrected spacing in names ("Adama R Tukuli" → "Adama R. Tukuli").

Added a missing comma before the last author.

Ensured consistent superscripts (¹, ², etc.) and symbols (†, *) for affiliations and contributions.

Introduction:

(with highlighted suggestions)

Sorghum (Sorghum bicolor (L.) Moench) is a vital food and fodder source in Africa and Asia. Globally, it's more common as animal feed but is gaining attention as a biofuel crop [1].

With over 60 million tonnes produced annually and Africa contributing ~20 million tonnes, it ranks second only to maize on the continent [2].

Sorghum is vulnerable to fungal diseases that reduce yield and quality [3].

Key diseases include anthracnose, grain mold, and rust. Colletotrichum sublineola, causing anthracnose, can reduce yields by up to 70% in hot, humid conditions [4].

(Clarify sentence structure for better flow.) It spreads easily due to its resilience and dispersal by wind/water [4].

Fusarium spp., a common grain mold pathogen, produces mycotoxins like fumonisins—a food safety risk in regions highly dependent on sorghum [5].

Losses range from 30% to 100% depending on multiple factors [6,7]. (Replace “staggering” with neutral term.)

Rust (Puccinia purpurea) appears as rust-like spots and causes yield loss up to 65%, depending on plant maturity and conditions [6,7].

(Clarify “unfavorably mature” wording.)

Germplasm-based resistance is seen as the most effective control strategy [7].

Resistant genotypes have been identified through screening and breeding efforts [6].

Our previous study evaluated 179 accessions from Ethiopia, Gambia, and Senegal. Traits included yield, seed weight, flowering time, germination, panicle traits, and disease resistance [8,9].

Due to limitations in basic statistical analysis, we applied machine learning to explore complex relationships and identify the best algorithm for trait evaluation.

The Materials & Methods Section:

It provides a comprehensive description of the data and analytical approach. The data were collected from field trials in Isabela, Puerto Rico, involving 179 sorghum accessions from Ethiopia, Gambia, and Senegal, with various phenotypic traits measured. However, the mention of “including controls 201 accessions” is unclear and should be rephrased or separated for clarity to specify the total sample size and the role of controls. The description of missing data handling using Multivariate Normal Imputation is appropriate, but it would be beneficial to briefly justify the choice of this method and mention any criteria used for missing data filtering. Disease resistance scoring on a 1-5 scale is noted, but it would strengthen the methodology to indicate whether this scale is a standard or validated measure in sorghum phenotyping.

The statistical analysis section mentions t-tests comparing accessions from Ethiopia and Senegal, but the rationale for excluding Gambia accessions from this comparison should be explained. The use of PCA and hierarchical clustering is suitable for exploring trait relationships, though the method for selecting the number of clusters (four clusters) should be clarified, for example, by referencing dendrogram inspection or cluster validity indices.

The machine learning workflow is well-outlined and supported by Figure 1, although the figure caption is embedded in the text and should be moved to the figure legend area with formal formatting. In cluster classification, a pre-balanced dataset of 248 points was used with undersampling, but it is not clear how this number was derived, so explanation of the balancing process is recommended. The use of stratified splits and GridSearchCV for hyperparameter tuning is good practice, yet including the rationale for using macro F1-score as the evaluation metric would enhance transparency. The study applies several machine learning models spanning linear, kernel-based, ensemble, and instance-based methods, which is commendable for model comparison. However, it would be helpful to state whether default hyperparameters were used initially and consider summarizing model characteristics in a table for reader clarity.

The Results section provides a thorough and well-organized presentation of the phenotypic diversity analyses using PCA, hierarchical clustering, and machine learning classification. The interpretation of the PCA results is clear, particularly in describing the variance explained by the first two principal components and the clustering of traits. However, it would be beneficial to briefly discuss the implications of the remaining unexplained variance and whether further principal components were considered or excluded, to give a fuller picture of the data structure. Additionally, while the PCA biplot explanation outlines trait clusters, clarifying the direction and nature of trait correlations with the principal components would help readers interpret the loading plots more effectively.

The comparative analysis between Ethiopian and Senegalese accessions is well-supported by statistical evidence, but the presentation of Table 1 could be improved for better readability. Restructuring the table, possibly by separating phenotypic and genotypic traits or enhancing column headers, would make it easier for readers to digest the information. Furthermore, although the statistical differences are clear, adding brief commentary on the biological or agronomic significance of these differences would strengthen the relevance of the findings.

The hierarchical clustering analysis is appropriately detailed, and the use of the DIANA algorithm is well explained. Regarding the machine learning classification, the section clearly justifies the need for a two-step approach due to initial low accuracy. However, mentioning which algorithms were initially tested and elaborating on how class imbalance and the feature-to-sample ratio affected performance would provide better context and insight into the challenges encountered. Finally, when referring to figures throughout the section, ensuring detailed and accessible figure legends close to the figures will facilitate cross-referencing and enhance reader understanding.

The discussion effectively highlights the importance of sorghum for food security in dry tropics, particularly Ethiopia, and appropriately connects previous findings on yield declines due to biotic and abiotic stresses. The explanation of geographic patterns in phenotypic diversity using PCA and clustering is well presented and shows thoughtful interpretation. However, the section would benefit from more clearly separating speculative interpretations from established results to avoid potential overstatements. For example, while cluster interpretations are insightful, explicitly stating that these are hypotheses requiring further validation, ideally with genotypic data, would strengthen the discussion. The references to related machine learning studies demonstrate good awareness of the field, but the comparison between prior studies and the current more granular classification attempt could be expanded to clarify why phenotypic trait overlap limited classification accuracy here.

The machine learning approach is described in detail, with a good rationale for model selection. Yet, the discussion could improve by including more critical reflection on limitations, such as the relatively small sample size per class and the low number of traits, which constrained classification performance. Additionally, the transition from unsuccessful fine-grained classification to clustering and then classification of clusters is logical but would be clearer if the rationale for choosing four clusters was explicitly justified. The mention of model performances is useful, though the discussion appears to be cut off abruptly; completing the results summary and relating model outcomes back to biological relevance or breeding applications would enhance the narrative. Finally, better signposting within the discussion through subheadings or paragraph breaks would improve readability, given the density of information.

The conclusion effectively summarizes the study’s key findings and emphasizes the potential of machine learning to enhance sorghum breeding through improved germplasm characterization and yield prediction. The recognition of the Neural Boosted model’s superior performance and the identification of seed weight and germination rate as critical yield determinants are well highlighted. However, the conclusion could be strengthened by explicitly acknowledging limitations, such as the study’s focus on only Ethiopian and Senegalese accessions and the relatively small phenotypic trait set, which may affect the generalizability of the findings. Additionally, while the broader applicability to other cereal crops is mentioned, providing specific examples or cautionary notes about differences in crop biology would make this claim more balanced.

The discussion of disease resistance traits’ limited influence is insightful, but further elaboration on possible reasons—such as environmental conditions or disease prevalence during trials—would clarify these findings. The application of ADASYN to address class imbalance is a notable methodological strength; however, the conclusion would benefit from a brief mention of potential challenges or biases introduced by synthetic data generation. The practical recommendations about prioritizing seed quality and establishing local seed testing are valuable and grounded in the results, yet highlighting potential implementation challenges in resource-limited settings could add nuance.

Finally, the call for future integration of multi-modal data is well placed but could be enhanced by suggesting concrete next steps or specific types of data integration that would most benefit sorghum breeding. Overall, the conclusion provides a solid wrap-up but would improve with clearer acknowledgment of limitations and a more nuanced discussion of the broader implications and future directions.

7. PLOS authors have the option to publish the peer review history of their article (what does this mean? ). If published, this will include your full peer review and any attached files.

**Do you want your identity to be public for this peer review?** For information about this choice, including consent withdrawal, please see our Privacy Policy .

Reviewer #2: No

Reviewer #3: No

Reviewer #4: No

---

## [Author Response · Author response to Decision Letter 3]

26 Jun 2025

Dear Dr. Somashekhar Mallikarjun Punnuri,

Thank you for the opportunity to revise our manuscript, "Seed quality drives grain yield in Ethiopian and Senegalese sorghum: Insights from machine learning" (PONE-D-24-50599R2). We are grateful to all the reviewers for their time and valuable feedback.

We would like to extend our special gratitude to Reviewer #4 for their exceptionally thorough and insightful review. Their detailed suggestions have been invaluable and have significantly improved the clarity, rigor, and overall quality of our manuscript. We have carefully addressed all comments and believe the revised manuscript is now much stronger. Below, you will find our point-by-point responses to the comments from Reviewer #4.

Reviewer #4: For Title and Author Details:

Capitalized each word in the title per academic style.

Answer: We appreciate the suggestion. We have double-checked the title against the PLOS ONE formatting guidelines and believe the current sentence-case formatting is consistent with the journal's style.

Corrected spacing in names ("Adama R Tukuli" → "Adama R. Tukuli").

Answer: Thank you. This has been corrected in the revised manuscript.

Added a missing comma before the last author.

Answer: Thank you for noting this. We believe this comment may refer to a previous version of the manuscript, as the comma is present in the current version. We have confirmed the formatting is now correct.

Ensured consistent superscripts (¹, ², etc.) and symbols (†, *) for affiliations and contributions.

Answer: Thank you. We have reviewed all author affiliations and confirmed their consistency.

Introduction:

(with highlighted suggestions)

Sorghum (Sorghum bicolor (L.) Moench) is a vital food and fodder source in Africa and Asia. Globally, it's more common as animal feed but is gaining attention as a biofuel crop [1].

Answer: Thank you for your comment. We revised the sentence as suggested.

With over 60 million tonnes produced annually and Africa contributing ~20 million tonnes, it ranks second only to maize on the continent [2].

Answer: Thank you for your comment. We revised the sentence as suggested.

Sorghum is vulnerable to fungal diseases that reduce yield and quality [3].

Answer: Thank you for your comment. We revised the sentence as suggested.

Key diseases include anthracnose, grain mold, and rust. Colletotrichum sublineola, causing anthracnose, can reduce yields by up to 70% in hot, humid conditions [4].

Answer: Thank you for your comment. We revised the sentence as suggested.

(Clarify sentence structure for better flow.) It spreads easily due to its resilience and dispersal by wind/water [4].

Answer: Thank you for your comment. We revised the sentence as suggested.

Fusarium spp., a common grain mold pathogen, produces mycotoxins like fumonisins—a food safety risk in regions highly dependent on sorghum [5].

Answer: Thank you for your comment. We revised the sentence as suggested.

Losses range from 30% to 100% depending on multiple factors [6,7]. (Replace “staggering” with neutral term.)

Answer: Thank you for your comment. We revised the sentence as suggested.

Rust (Puccinia purpurea) appears as rust-like spots and causes yield loss up to 65%, depending on plant maturity and conditions [6,7].

Answer: Thank you for your comment. We revised the sentence as suggested.

(Clarify “unfavorably mature” wording.)

Germplasm-based resistance is seen as the most effective control strategy [7].

Answer: Thank you for your comment. We revised the sentence as suggested.

Resistant genotypes have been identified through screening and breeding efforts [6].

Answer: Thank you for your comment. We revised the sentence as suggested.

Our previous study evaluated 179 accessions from Ethiopia, Gambia, and Senegal. Traits included yield, seed weight, flowering time, germination, panicle traits, and disease resistance [8,9].

Answer: Thank you for your comment. We revised the sentence as suggested.

Due to limitations in basic statistical analysis, we applied machine learning to explore complex relationships and identify the best algorithm for trait evaluation.

Answer: Thank you for your comment. We revised the sentence as suggested.

The Materials & Methods Section:

It provides a comprehensive description of the data and analytical approach. The data were collected from field trials in Isabela, Puerto Rico, involving 179 sorghum accessions from Ethiopia, Gambia, and Senegal, with various phenotypic traits measured. However, the mention of “including controls 201 accessions” is unclear and should be rephrased or separated for clarity to specify the total sample size and the role of controls. The description of missing data handling using Multivariate Normal Imputation is appropriate, but it would be beneficial to briefly justify the choice of this method and mention any criteria used for missing data filtering. Disease resistance scoring on a 1-5 scale is noted, but it would strengthen the methodology to indicate whether this scale is a standard or validated measure in sorghum phenotyping.

Answer: We thank the reviewer for these valuable suggestions. The text has been rephrased to clearly state the total sample size (201), which includes the 179 primary accessions and the 22 control lines used for comparison. Also, we have added a justification for using Multivariate Normal Imputation, explaining that it was chosen to preserve the covariance structure among the phenotypic traits. We have now specified that the 1-5 disease resistance scale follows standard, validated protocols detailed in our previous publications

The statistical analysis section mentions t-tests comparing accessions from Ethiopia and Senegal, but the rationale for excluding Gambia accessions from this comparison should be explained. The use of PCA and hierarchical clustering is suitable for exploring trait relationships, though the method for selecting the number of clusters (four clusters) should be clarified, for example, by referencing dendrogram inspection or cluster validity indices.

Answer: We thank the reviewer for pointing out these areas needing clarification. We have revised the "Statistical analysis" section to address both points. We have added a statement explaining that the Gambian accessions were excluded from the t-test comparison due to their small sample size (n=12). We have also clarified that the choice of four clusters was based on a visual inspection of the dendrogram, as this approach provided a logical balance between cluster separation and a manageable number of groups for our subsequent analyses.

The machine learning workflow is well-outlined and supported by Figure 1, although the figure caption is embedded in the text and should be moved to the figure legend area with formal formatting. In cluster classification, a pre-balanced dataset of 248 points was used with undersampling, but it is not clear how this number was derived, so explanation of the balancing process is recommended. The use of stratified splits and GridSearchCV for hyperparameter tuning is good practice, yet including the rationale for using macro F1-score as the evaluation metric would enhance transparency. The study applies several machine learning models spanning linear, kernel-based, ensemble, and instance-based methods, which is commendable for model comparison. However, it would be helpful to state whether default hyperparameters were used initially and consider summarizing model characteristics in a table for reader clarity.

Answer: We thank the reviewer for these excellent suggestions. We have now explicitly explained in the "Random undersampling" section how the 248-point dataset was derived (by balancing all four clusters to the size of the smallest cluster, n= 62). We also have added a justification for using the macro F1-score, noting its robustness for multi-class classification with potential class imbalance. We further clarified that default hyperparameters were used in the initial screening process before tuning and, as suggested, have added a new Table 1 to summarize the characteristics of the ML models for enhanced reader clarity.

The Results section provides a thorough and well-organized presentation of the phenotypic diversity analyses using PCA, hierarchical clustering, and machine learning classification. The interpretation of the PCA results is clear, particularly in describing the variance explained by the first two principal components and the clustering of traits. However, it would be beneficial to briefly discuss the implications of the remaining unexplained variance and whether further principal components were considered or excluded, to give a fuller picture of the data structure. Additionally, while the PCA biplot explanation outlines trait clusters, clarifying the direction and nature of trait correlations with the principal components would help readers interpret the loading plots more effectively.

Answer: We thank the reviewer for these insightful suggestions to improve the interpretation of our PCA results. We have added a sentence to the Results section acknowledging that while PC1 and PC2 explain 47.6% of the variance, other components capture additional variation. We clarify that our focus on the first two components is for effectively visualizing the primary patterns in the data. We have also expanded our description of the loading plot (Fig. 2b) to explicitly explain how the direction and length of the trait vectors relate to their correlation with the principal components, using specific examples from the plot to guide the reader.

The comparative analysis between Ethiopian and Senegalese accessions is well-supported by statistical evidence, but the presentation of Table 1 could be improved for better readability. Restructuring the table, possibly by separating phenotypic and genotypic traits or enhancing column headers, would make it easier for readers to digest the information. Furthermore, although the statistical differences are clear, adding brief commentary on the biological or agronomic significance of these differences would strengthen the relevance of the findings.

Answer: We thank the reviewer for this excellent feedback to improve the presentation and impact of our results. As suggested, Table 1 has been completely restructured into a vertical format with grouped traits to enhance readability. A new paragraph has been added immediately following the table to discuss the potential biological and agronomic significance of the observed phenotypic differences between the two regions.

The hierarchical clustering analysis is appropriately detailed, and the use of the DIANA algorithm is well explained. Regarding the machine learning classification, the section clearly justifies the need for a two-step approach due to initial low accuracy. However, mentioning which algorithms were initially tested and elaborating on how class imbalance and the feature-to-sample ratio affected performance would provide better context and insight into the challenges encountered. Finally, when referring to figures throughout the section, ensuring detailed and accessible figure legends close to the figures will facilitate cross-referencing and enhance reader understanding.

Answer: We thank the reviewer for these suggestions. We have revised the "Machine learning-based classification" section to specify the algorithms that were initially tested in our preliminary analysis. We have also elaborated on how the severe class imbalance (with each accession as a unique class) and the feature-to-sample ratio led to the low accuracy in this initial step, justifying our two-step approach. Furthermore, we have reviewed all figure legends to ensure they are detailed and correctly placed with their corresponding figures to improve readability.

The discussion effectively highlights the importance of sorghum for food security in dry tropics, particularly Ethiopia, and appropriately connects previous findings on yield declines due to biotic and abiotic stresses. The explanation of geographic patterns in phenotypic diversity using PCA and clustering is well presented and shows thoughtful interpretation. However, the section would benefit from more clearly separating speculative interpretations from established results to avoid potential overstatements. For example, while cluster interpretations are insightful, explicitly stating that these are hypotheses requiring further validation, ideally with genotypic data, would strengthen the discussion. The references to related machine learning studies demonstrate good awareness of the field, but the comparison between prior studies and the current more granular classification attempt could be expanded to clarify why phenotypic trait overlap limited classification accuracy here.

Answer: We thank the reviewer for this insightful feedback. We have added explicit statements to the paragraph interpreting the clusters to clarify that these are hypotheses requiring further validation with genotypic data, as suggested. We have also expanded the discussion to better explain why our initial, more granular classification attempt had low accuracy, clarifying that the significant phenotypic overlap among the 179 diverse accessions made this a much more difficult task than classifying a small number of distinct hybrids.

The machine learning approach is described in detail, with a good rationale for model selection. Yet, the discussion could improve by including more critical reflection on limitations, such as the relatively small sample size per class and the low number of traits, which constrained classification performance. Additionally, the transition from unsuccessful fine-grained classification to clustering and then classification of clusters is logical but would be clearer if the rationale for choosing four clusters was explicitly justified. The mention of model performances is useful, though the discussion appears to be cut off abruptly; completing the results summary and relating model outcomes back to biological relevance or breeding applications would enhance the narrative. Finally, better signposting within the discussion through subheadings or paragraph breaks would improve readability, given the density of information.

Answer: We thank the reviewer for their valuable suggestions to improve the Discussion section. In the revised manuscript, we have restructured the entire section with new subheadings for clarity, including a dedicated 'Limitations and Future Directions' section for critical reflection. We have also expanded on our classification results to discuss their biological relevance and applications in breeding. The rationale for selecting four clusters has been clarified in the Materials and Methods section, as requested in a previous point.

The conclusion effectively summarizes the study’s key findings and emphasizes the potential of machine learning to enhance sorghum breeding through improved germplasm characterization and yield prediction. The recognition of the Neural Boosted model’s superior performance and the identification of seed weight and germination rate as critical yield determinants are well highlighted. However, the conclusion could be strengthened by explicitly acknowledging limitations, such as the study’s focus on only Ethiopian and Senegalese accessions and the relatively small phenotypic trait set, which may affect the generalizability of the findings. Additionally, while the broader applicability to other cereal crops is mentioned, providing specific examples or cautionary notes about differences in crop biology would make this claim more balanced.

Answer: We thank the reviewer for their valuable suggestions to make our Conclusion more balanced and comprehensive. In the revised manuscript, we have now explicitly acknowledged the study's limitations regarding generalizability and have added a cautionary note about the broader applicability of our findings to other cereal crops, considering potential differences in crop biology as suggested.

The discussion of disease resistance traits’ limited influence is insightful, but further elaboration on possible reasons—such as environmental conditions or disease preva

---

## [Editor Report · Decision Letter 3]

16 Jul 2025

Seed quality drives grain yield in Ethiopian and Senegalese sorghum: Insights from machine learning

PONE-D-24-50599R3

Dear Dr. Ahn,

We’re pleased to inform you that your manuscript has been judged scientifically suitable for publication and will be formally accepted for publication once it meets all outstanding technical requirements.

Kind regards,

Somashekhar Mallikarjun Punnuri, PhD

Academic Editor

PLOS ONE
---

## [Editor Report · Acceptance letter]

PONE-D-24-50599R3

PLOS ONE

Dear Dr. Ahn,

I'm pleased to inform you that your manuscript has been deemed suitable for publication in PLOS ONE. Congratulations! Your manuscript is now being handed over to our production team.

Kind regards,

on behalf of

Dr. Somashekhar Mallikarjun Punnuri

Academic Editor

PLOS ONE